# Radical and lunatic fringes modulate notch ligands to support mammalian intestinal homeostasis

Preetish Kadur Lakshminarasimha Murthy[1,2†], Tara Srinivasan[3†],
Matthew S Bochter[4], Rui Xi[1], Anastasia Kristine Varanko[3], Kuei-Ling Tung[1,5],
Fatih Semerci[6], Keli Xu[7], Mirjana Maletic-Savatic[6], Susan E Cole[4], Xiling Shen[1,3,8]*

[1]Center for Genomics and Computational Biology, Department of Biomedical Engineering, Duke University, Durham, United States; [2]Sibley School of Mechanical and Aerospace Engineering, Cornell University, Ithaca, United States; [3]Mienig School of Biomedical Engineering, Cornell University, Ithaca, United States; [4]Department of Molecular Genetics, Ohio State University, Columbus, United States; [5]Department of Biological and Environmental Engineering, Cornell University, Ithaca, United States; [6]Department of Pediatrics, Baylor College of Medicine, Houston, United States; [7]Department of Neurobiology and Anatomical Sciences, University of Mississippi Medical Center, Jackson, United States; [8]School of Electrical and Computer Engineering, Cornell University, Ithaca, United States

*For correspondence:
xiling.shen@duke.edu

†These authors contributed equally to this work

Competing interests: The authors declare that no competing interests exist.

**Abstract** Notch signalling maintains stem cell regeneration at the mouse intestinal crypt base and balances the absorptive and secretory lineages in the upper crypt and villus. Here we report the role of Fringe family of glycosyltransferases in modulating Notch activity in the two compartments. At the crypt base, RFNG is enriched in the Paneth cells and increases cell surface expression of DLL1 and DLL4. This promotes Notch activity in the neighbouring *Lgr5+* stem cells assisting their self-renewal. Expressed by various secretory cells in the upper crypt and villus, LFNG promotes DLL surface expression and suppresses the secretory lineage . Hence, in the intestinal epithelium, Fringes are present in the ligand-presenting 'sender' secretory cells and promote Notch activity in the neighbouring 'receiver' cells. Fringes thereby provide for targeted modulation of Notch activity and thus the cell fate in the stem cell zone, or the upper crypt and villus.
DOI: https://doi.org/10.7554/eLife.35710.001

## Introduction

*Lgr5+* Crypt Base Columnar cells (CBCs) located at the bottom of the crypts constantly self-renew to maintain the small intestinal epithelium, which is one of the fastest regenerative tissues in the body (*Barker et al., 2007*; *van der Flier et al., 2009*). CBCs divide and move up the crypt into the progenitor or transit-amplifying zone where the cells rapidly proliferate and terminally differentiate into two major types: absorptive (enterocytes) and secretory (mainly Paneth and goblet cells). Enterocytes and goblet cells populate the villi while the Paneth cells move to the bottom of the crypt and provide a niche for the CBCs (*van der Flier et al., 2009*).

Notch signalling pathway primarily consists of Notch receptors (NOTCH1-4) and ligands (DLL1-4 and JAG1-2) (*Bray, 2006*). Upon activation of a Notch receptor by a ligand, it undergoes successive cleavages by ADAM/TACE and γ-secretase (*Bray, 2006*). The cleaved intracellular domain (NICD) translocates to the nucleus leading to the transcription of multiple genes such as *Hes* and *Hey* families (*Kopan, 2002*; *Iso et al., 2003*). The extracellular domain of the Notch receptor and ligands

contain EGF-like repeats, some of which serve as substrates for O-fucosylation by POFUT1 (*Rampal et al., 2007*; *Wang et al., 2001*). The fucosylated product may be further modified within the Golgi network by Fringe proteins: Lunatic (LFNG), Manic (MFNG) and Radical Fringe (RFNG) (*Moloney et al., 2000*; *Haines and Irvine, 2003*). Fringe proteins are typically expressed in receptor-expressing 'receiver' cells (*Haines and Irvine, 2003*). Glycosylation of NOTCH1 by LFNG and MFNG increases its activation by DLL1 but decreases its activation by JAG1 (*Haines and Irvine, 2003*; *Hicks et al., 2000*; *Panin et al., 1997*). In contrast, glycosylation by RFNG increases the activation of NOTCH1 by both DLL1 and JAG1 (*LeBon et al., 2014*).

Notch pathway provides for spatial and context specific decision making in the intestinal epithelium. At the bottom of the crypt, Notch signalling is important for the maintenance of CBCs (*Pellegrinet et al., 2011*). In the upper crypt however, Notch activity, mainly through *Hes1*, is essential for the enterocyte differentiation (*van der Flier et al., 2009*; *Fre et al., 2005*). Inhibition of Notch signalling results in the loss of proliferative CBCs and progenitor cells and leads to their differentiation into goblet cells in the upper crypt and villus, indicating the importance of the pathway in maintaining the tissue (*Pellegrinet et al., 2011*; *VanDussen et al., 2012*; *Riccio et al., 2008*; *Wu et al., 2010*). *Notch1, 2* and *Dll1, 4* are known to be the necessary receptors and ligands in the intestine (*Pellegrinet et al., 2011*; *Riccio et al., 2008*; *Schröder and Gossler, 2002*). Although, the fringe proteins are known to be expressed in the intestine, their function has not been studied (*Schröder and Gossler, 2002*).

Here we show that *Rfng* and *Lfng* are expressed by the ligand-presenting secretory lineages, but at different locations. At the crypt base, *Rfng* expressed in Paneth cells modulates DLL1 and DLL4, which enhances Notch signalling and self-renewal of neighbouring CBCs. In the upper crypt and villus, *Lfng* is expressed by secretory cells including enteroendocrine, Tuft and goblet cells. LFNG promotes Notch signalling in the transit amplifying cells and impedes their differentiation into secretory cells. MFNG does not play any noticeable role in intestinal epithelial homeostasis.

## Results

### *Rfng* supports *Lgr5+* stem cell self-renewal

*Rfng* transcripts have been detected in the crypt by in situ hybridisation (*Schröder and Gossler, 2002*). We analysed previously published microarray data on *Lgr5+* CBCs and Paneth cells (*Sato et al., 2011*) and found that *Rfng* is significantly upregulated in Paneth cells (*Figure 1—figure supplement 1A*). We isolated CBCs and Paneth cells (CD24$^{high}$/SSC$^{high}$) from *Lgr5*-GFP mice by FACS using an established protocol (*Sato et al., 2011*; *Sato et al., 2009*) and found that the Paneth cells are enriched for *Rfng* (*Figure 1A*). We validated that the isolated cells are indeed Paneth cells and CBCs by confirming their Lysozyme and GFP expression respectively (*Figure 1—figure supplement 1B,C*). We also confirmed that *Rfng* is enriched in the Paneth cells by RNA in situ hybridisation (ISH) (*Figure 1B*). We validated the specificity of ISH probes using *Rfng* null mouse intestinal sections (*Figure 1—figure supplement 1D,E*).

We then established an in vitro knockdown (KD) model using organoid cultures of epithelial cells obtained from *Lgr5*-GFP mice. Single *Lgr5*-GFP+ CBCs were transduced with scrambled (Sc.) shRNA (control) or *Rfng* shRNA and propagated as organoids (*Figure 1—figure supplement 1F*). The colony formation efficiency of the *Rfng* KD CBCs was reduced compared to the control (*Figure 1C*). Flow cytometric analysis showed that the number of *Lgr5+* CBCs decreased after *Rfng* loss, whereas the number of Paneth cells remained relatively unchanged (*Figure 1D*).

We confirmed the observation in vivo using previously published *Rfng* deficient (*Rfng$^{-/-}$*) mice (*Moran et al., 2009*). Crypt cells were isolated from *Rfng$^{-/-}$* and *Rfng$^{+/+}$* control mice and analysed by flow cytometry using a combination of surface markers to identify CBCs (CD24$^{lo}$CD44$^+$CD166$^+$-GRP78$^-$) (*Wang et al., 2013*). Analysis revealed that the *Rfng$^{-/-}$* mice had almost a two-fold depletion in CBCs (*Figure 1E*). A reduction of CBC marker *Lgr5* transcripts in the crypts harvested from *Rfng$^{-/-}$* mouse intestines was observed by RT-qPCR measurement when compared to the control (*Figure 1F*). The number of Paneth and goblet cells remain largely unchanged and no other significant phenotype was detected in the epithelium (*Figure 1—figure supplement 2A–F*). Loss of *Rfng* in organoids seems to show a more significant phenotype in CBC reduction than its loss in vivo. This

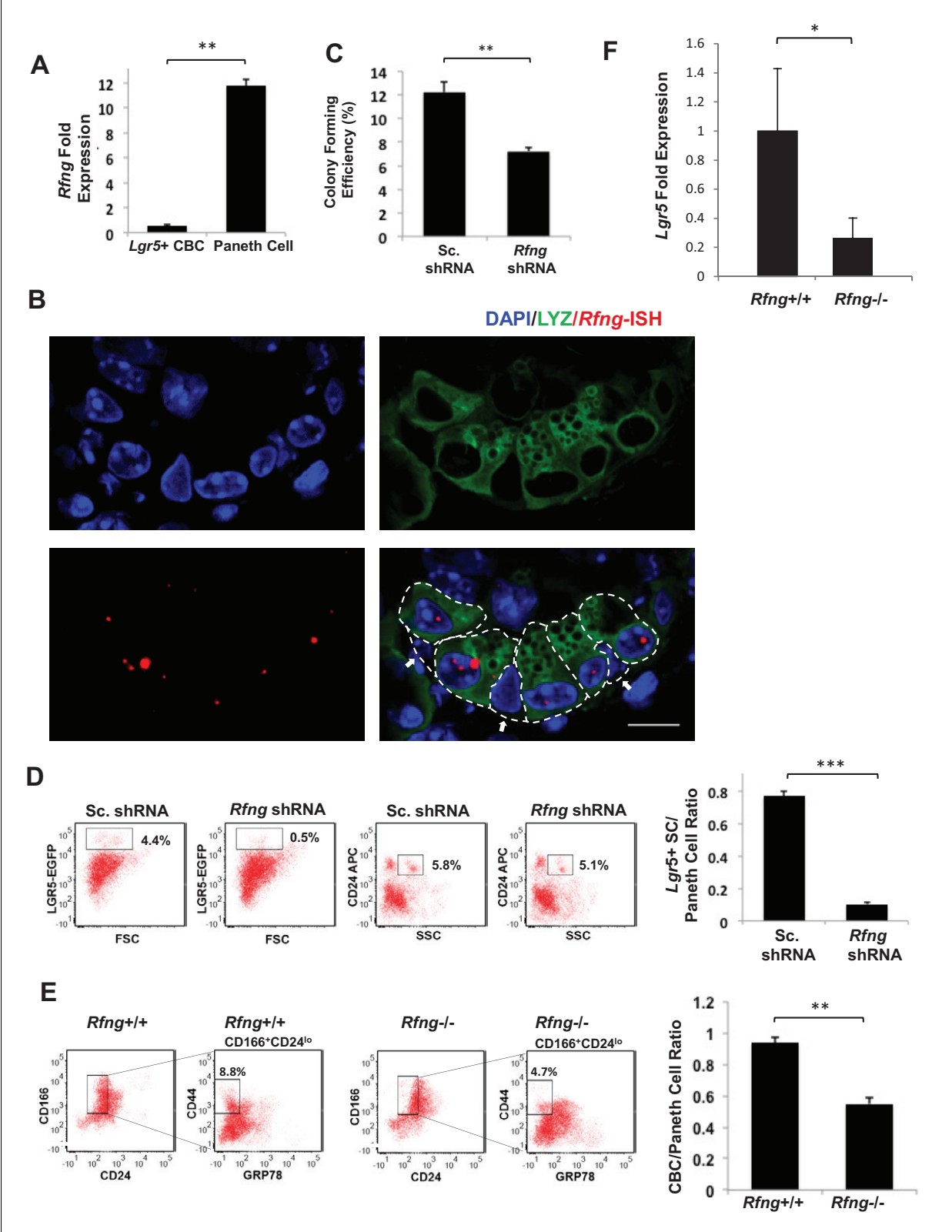

**Figure 1.** *Rfng* supports *Lgr5*+ stem cell self-renewal. (**A**) RT-qPCR quantification of *Rfng* in *Lgr5*+ CBC and Paneth cells isolated from *Lgr5*-GFP mouse intestines. The experiment was performed in triplicate and presented as mean ± s.d. (standard deviation) (**B**) Representative image showing *Rfng* transcripts (red) and Lysozyme protein (green) expression at the bottom of the crypt of *Lgr5*-GFP mouse intestine. DAPI (Blue) labels the nuclei and scale bar represents 10 μm. Arrows point to CBCs. (**C–D**) Single *Lgr5*-GFP CBCs were transduced with either Sc. shRNA or *Rfng* shRNA. The experiment

*Figure 1 continued on next page*

Figure 1 continued

was performed in triplicate. (C) Colony forming efficiency measured after 7 days. Quantitative analysis calculated from 1000 cells/replicate presented as mean ± s.d. (D) Left: Representative flow cytometry plots indicating gated percentage of *Lgr5*+ (GFP$^{high}$) and Paneth cells (CD24$^{high}$/SSC$^{high}$). Right: Ratio of *Lgr5*-GFP+ CBCs/Paneth cells as determined by flow cytometry and presented as mean ± s.d. (E) Left: Representative plots indicating gated population of CBCs (CD166$^+$CD24$^{lo}$CD44$^+$GRP78$^-$) from the intestine of *Rfng$^{+/+}$* and *Rfng$^{-/-}$* mice. Percentage reflects fraction of total population. Right: Ratio of number of CBCs to Paneth cells of n = 3 mice and presented as mean ± s.d. (F) RT-qPCR quantification of *Lgr5* in crypts extracted from *Rfng$^{+/+}$* and *Rfng$^{-/-}$* mice. n = 3 mice. Data is presented as mean ± s.d. (*p<0.05; **p<0.01; ***p<0.001).

DOI: https://doi.org/10.7554/eLife.35710.002

The following figure supplements are available for figure 1:

**Figure supplement 1.** Paneth Cells express *Rfng*.

DOI: https://doi.org/10.7554/eLife.35710.003

**Figure supplement 2.** Histological and flow cytometric analysis of *Rfng* null intestines.

DOI: https://doi.org/10.7554/eLife.35710.004

**Figure supplement 3.** Colony formation ability of *Lgr5*+ CBCs when mixed with Paneth cells from control or *Rfng* null mice.

DOI: https://doi.org/10.7554/eLife.35710.005

may be because CBCs in vivo also receive cues from the mesenchyme and not just the Paneth cells as in case of organoids.

To confirm that the loss of *Rfng* only in the Paneth cells can affect the CBCs, we performed the Organoid Reconstitution Assay (ORA) described previously (*Rodríguez-Colman et al., 2017*). FACS sorted *Lgr5*-GFP+ CBCs were incubated with Paneth cells from wild type or *Rfng* null mice for 10 min at room temperature and plated in Matrigel. We find that the colony formation ability of CBCs incubated with Paneth cells lacking *Rfng* was significantly lower than the control (*Figure 1—figure supplement 3*). It should be noted that not all CBCs associate with a Paneth cell during the incubation. Also, the *Lgr5*-GFP+ CBCs divide and give rise to Paneth cells with *Rfng*. Hence the result of this assay is not as significant as that shown in *Figure 1C*.

## *Rfng* promotes Notch signalling in CBCs

We isolate, by FACS, the CBCs and Paneth cells from the *Rfng* KD and control seven days old organoids described earlier. Western blotting shows that Notch target genes *Hes1* and *Hey1* have reduced expression in the CBCs upon loss of *Rfng*. However, the levels of ligands DLL1, four and JAG1 on Paneth cells were not significantly altered, consistent with the post-translational modifying role of Fringe (*Figure 2A*). *Rfng* KD and control CBCs were then transfected with an RBPJκ-dsRed reporter (*Hansson et al., 2006*) as an indicator of Notch activity, cultured overnight and analysed by flow cytometry (*Figure 2B*). *Rfng* shRNA decreased the mean RBPJκ-dsRed fluorescent intensity, indicating reduced overall Notch signalling in CBCs.

Fringes are known to modulate Notch signalling when present in receptor expressing cell (*Haines and Irvine, 2003*). But here we find *Rfng* in the ligand presenting cell promoting Notch activity in the neighbouring CBCs. We confirmed that the Paneth cells express the ligand *Dll1* by RNA-ISH (*Figure 2—figure supplement 1*). To understand the mechanism behind this, we examined ligand availability and concentration on the cell surface according to a previously established method using flow cytometry (*LeBon et al., 2014*; *Taylor et al., 2014*; *Yang et al., 2005*). Seven days old *Rfng* KD and control organoids were dissociated and single unpermeabilised cells were labelled with CD24 to mark Paneth cells and NOTCH1-Fc to quantify ligand binding to NOTCH1 (*Figure 2C*). Mean fluorescent intensity (MFI) of NOTCH1 binding to Paneth cells with *Rfng* knockdown was reduced compared to the scrambled control. We further confirmed that the ligands available on the Paneth cell surface have reduced by using specific antibodies for DLL1, DLL4 and JAG1. Flow cytometry showed that DLL1 and DLL4 levels on the Paneth cell surface reduced after the loss of *Rfng* (*Figure 2D*), although the total expression level of DLL1 and DLL4 within Paneth cells is not changed by *Rfng* knockdown after the cells were permeabilised (*Figure 2E*). Western blotting also confirmed that the total DLL1 and DLL4 expression in Paneth cells does not change significantly after the loss of *Rfng* (*Figure 2A*). The outcomes of the ligand availability assays suggest that the DLLs available on the Paneth cell surface for NOTCH1 to bind to have been reduced after the loss of RFNG, which could decrease Notch activity in the adjacent CBCs.

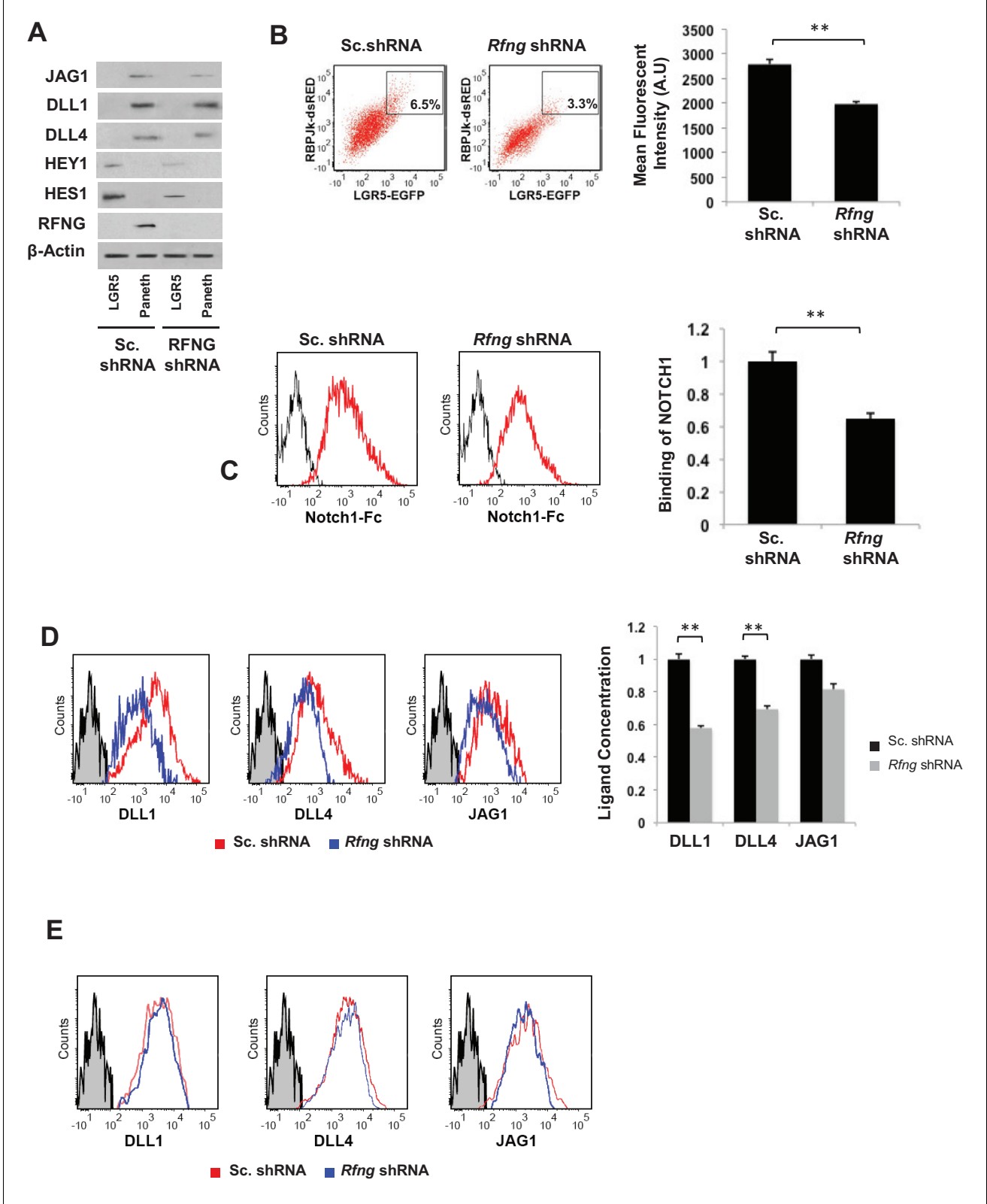

**Figure 2.** *Rfng* promotes Notch signaling in *Lgr5*+ CBC. (**A**) Western blot analysis of Notch signaling components in CBCs and Paneth cells FACS sorted from *Rfng* KD and control organoids. (**B**) Left: Representative plots for RBPJκ-dsRed and *Lgr5*-GFP expression indicating a gated double positive fraction for *Rfng* KD and control CBCs transfected with RBPJκ-dsRed reporter. Right: Mean fluorescence intensity (MFI) of RBPJκ-dsRed expression. The experiment was performed in triplicate and presented as mean ± s.d. (**C**) Ligand availability on *Rfng* KD and control Paneth cells. Representative traces

*Figure 2 continued on next page*

*Figure 2 continued*

(left) and MFI (right) showing ligand binding to NOTCH1 measured by flow cytometry. Unstained Paneth cells were used as a negative control. The experiment was performed in triplicate and presented as mean ± s.d. (D) Cell surface DLL1, DLL4, and JAG1 concentration on *Rfng* KD and control unpermeabilised Paneth cells. Left: Representative traces measured by flow cytometry. Right: MFI measurements. The experiment was performed in triplicate and presented as mean ± s.d. (E) Cell surface DLL1, DLL4, and JAG1 concentration on *Rfng* KD and control permeabilised Paneth cells. (**p<0.01).

DOI: https://doi.org/10.7554/eLife.35710.006

The following figure supplement is available for figure 2:

**Figure supplement 1.** *Dll1* expression in the crypts.

DOI: https://doi.org/10.7554/eLife.35710.007

## *Mfng* plays an insignificant role

*Mfng* is expressed by scattered cells in the intestinal epithelium (*Schröder and Gossler, 2002*). To understand its potential function in maintaining the epithelium, we established an in vitro shRNA based *Mfng* knockdown model as before. Western blotting and RT-qPCR analysis validated *Mfng* knockdown (*Figure 3A,B*). Gene expression levels of *Lgr5* and Notch components were comparable between Sc. Control and *Mfng* KD organoids (*Figure 3B*). Additionally, the colony forming efficiency (*Figure 3C*) and the expression pattern of *Lgr5*-GFP (CBC) and MUC2 (goblet cells) (*Figure 3D*) of *Mfng* shRNA-expressing CBCs were similar to the scrambled control. We quantified this observation using flow cytometry, which confirmed no significant change in the number of *Lgr5*-GFP+ CBCs and goblet cells (*Figure 3E,F*). Finally, the percentage of differentiated cells, identified by CK20 expression, was not significantly altered between Sc. control and *Mfng* KD organoids (*Figure 3G*).

We then analysed intestinal tissues from *Mfng* deficient (*Mfng*$^{-/-}$) mice (*Moran et al., 2009*) (*Figure 3—figure supplement 1A–C*). IF microscopy showed similar MUC2 staining in intestinal sections of *Mfng*$^{-/-}$ and wild-type (*Mfng*$^{+/+}$) mice (*Figure 3—figure supplement 1D*). Quantification in intestinal tissues based on IF expression indicated that the number of goblet cells was not significantly altered in *Mfng*$^{+/+}$ mice compared to *Mfng*$^{-/-}$ mice. Finally, we examined the total number of CK20+ cells in intestines, which was similar in wild-type and *Mfng* null mice (*Figure 3—figure supplement 2A*).

We observed that goblet cells were slightly enriched in *Mfng* when compared to the CBCs (*Figure 3—figure supplement 2B*). We found no significant change in the cell surface expression of DLLs on goblet cells after the loss of *Mfng* (*Figure 3—figure supplement 2C–F*). Overall, these data suggest *Mfng* plays an insignificant role in intestinal tissues.

## *Lfng* deletion leads to increased goblet cell differentiation

Lunatic Fringe is known to be expressed in the crypts and scattered cells in the villous epithelium (*Schröder and Gossler, 2002*). Immunofluorescence analysis of intestines from *Lfng*-GFP reporter mice confirmed that *Lfng* is expressed by a subset of cells in the upper crypt (transient-amplifying cell region) and villus (*Figure 4A*). We observe that *Lfng*-GFP+ cells are post-mitotic in the upper-crypt. (*Figure 4A*). Further analysis showed that these *Lfng*-GFP+ cells express *ChgA*, *Dclk1* or *Muc2* which are markers for enteroendocrine, Tuft or goblet cells respectively (*Figure 4B–D*). Secretory cells in the intestine, mainly enteroendocrine and goblet cells, are known to express Notch ligands, especially DLL1 (*van Es et al., 2012*). In the upper crypt, immunofluorescence analysis showed Notch1 activity in the cells adjacent to *Lfng*-GFP+ cells but not in themselves (*Figure 4E*). We performed RT-qPCR measurement of *Lfng* using goblet cells and CBCs isolated from *Lgr5*-GFP mice and confirmed that *Lfng* is in goblet cells and not CBCs (*Figure 4—figure supplement 1A*).

We established an in vitro shRNA based *Lfng* knockdown model as before (*Figure 4—figure supplement 1B*). We observed only a slight decrease in colony forming efficiency of CBCs after *Lfng* knockdown and no significant change in the level of Notch activity in the CBCs (*Figure 4—figure supplement 1C,D*). However, we find that the number of goblet cells (MUC2+) increased after the loss of *Lfng* (*Figure 4C*). Quantification by flow cytometry showed that number of goblet cells (UEA-1+/CD24-) (*Wong et al., 2012*) was increased significantly in *Lfng* KD organoids (5.5% of the total population) when compared to the scrambled control (1.9%) (*Figure 4D*). Accordingly, the ratio of

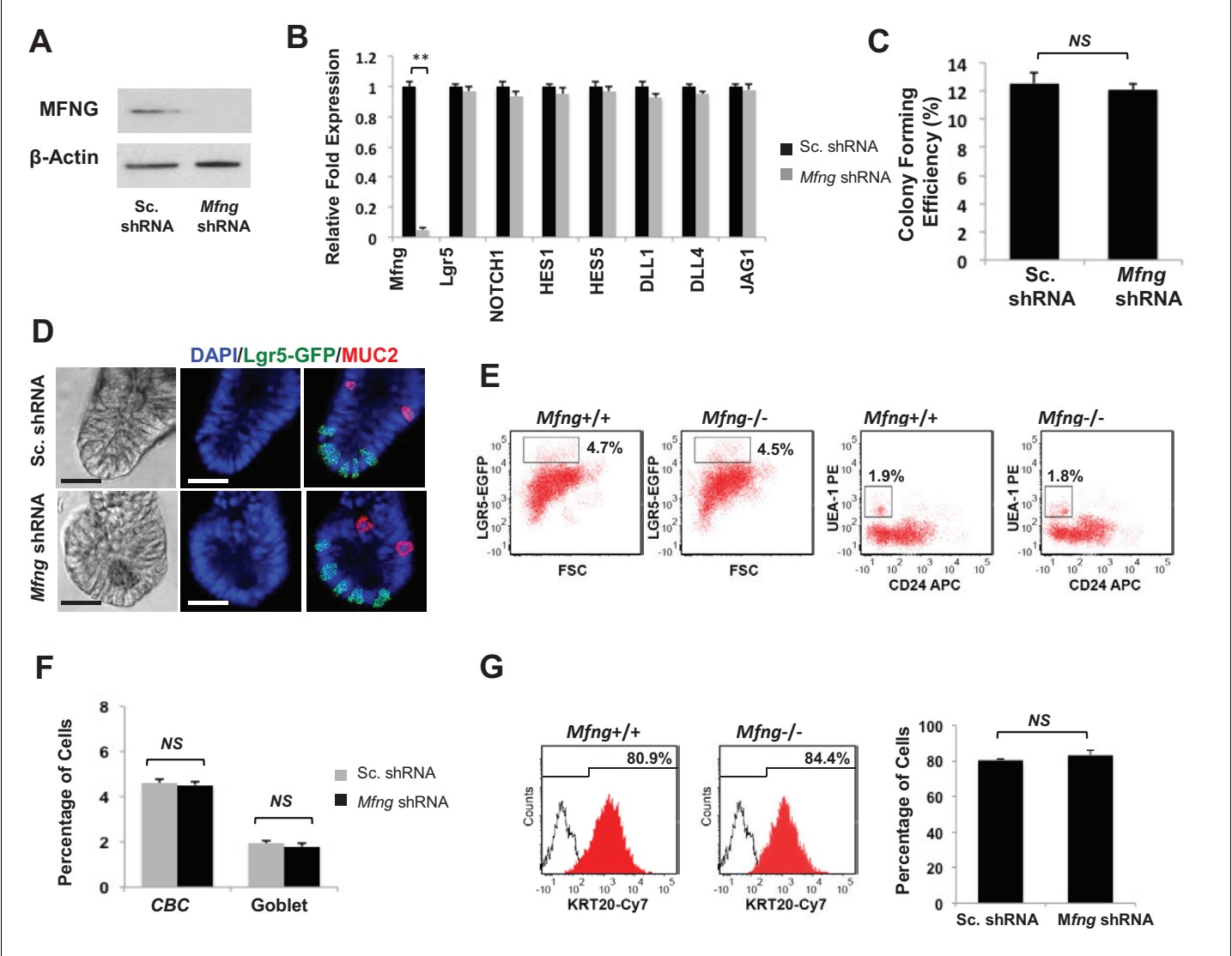

**Figure 3.** *Mfng* plays an insignificant role. Single *Lgr5*-GFP CBCs were transduced with either Sc. shRNA or *Mfng* shRNA. The experiment was performed in triplicate. (**A**) Western blot for *Mfng* expression. (**B**) RT-qPCR quantification of *Mfng* and Notch components in organoids. (**C**) Colony forming efficiency measured after 7 days. Quantitative analysis from 1000 cells/replicate. (**D**) Representative bright field and co-IF images indicating *Lgr5*-GFP (green) expression. MUC2 (red) marks Goblet cells. DAPI (blue) labels nuclei and scale bar represents 25 μm. (**E**) Representative flow cytometry plots indicating gated percentage of *Lgr5*+ CBCs (GFP[high]) and goblet cells (UEA-1[+]/CD24[-]). (**F**) Percentage of *Lgr5*+ CBCs and goblet cells as determined by flow cytometry and presented as mean ± s.d. (**G**) Left: Representative flow cytometry histograms indicating KRT20+ (CK20+) cells. Right: Percentage of KRT20+ cells and presented as mean ± s.d.

DOI: https://doi.org/10.7554/eLife.35710.008

The following figure supplements are available for figure 3:

**Figure supplement 1.** Histological analysis of *Mfng* null intestines.
DOI: https://doi.org/10.7554/eLife.35710.009

**Figure supplement 2.** No significant phenotype detected upon loss of Mfng.
DOI: https://doi.org/10.7554/eLife.35710.010

the number of goblet cells to *Lgr5*+ CBCs increased approximately three times in *Lfng* shRNA-expressing organoids (*Figure 4E*).

We confirmed these observations in vivo by examining the intestinal tissues from *Lfng* deficient (*Lfng*$^{-/-}$) mice (*Moran et al., 2009*). We observed an increase in the number of goblet cells in the *Lfng* null mice as expected (*Figure 5A* and *Figure 5—figure supplement 1A,B*). Goblet cells were

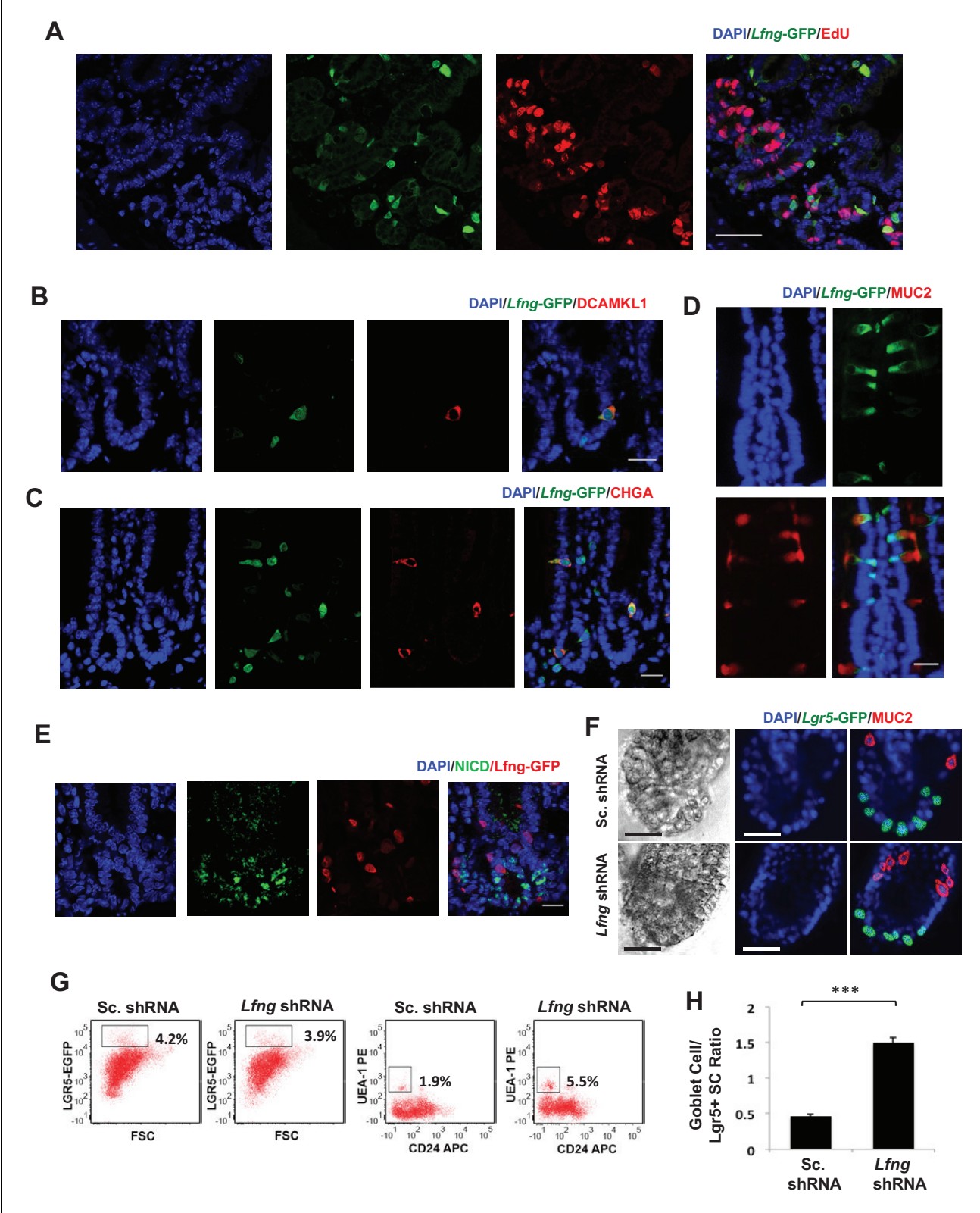

**Figure 4.** *Lfng* loss results in increased goblet cell differentiation in vitro. (**A–E**) Representative IF images of the small intestine of *Lfng*-GFP reporter mice. (**A**) GFP (green) shows the *Lfng* expression and EdU (red) marks the proliferating cells. DAPI (blue) labels nuclei. Scale bar represents 50 μm. (**B**) GFP (green) shows the *Lfng* expression and DCAMKL1 (red) marks the Tuft cells. Scale bar represents 20 μm. (**C**) GFP (green) shows the *Lfng* expression and CHGA (red) marks the enteroendocrine cells. Scale bar represents 20 μm. (**D**) GFP (green) shows the *Lfng* expression and MUC2 (red) marks the

*Figure 4 continued on next page*

Figure 4 continued

goblet cells. Scale bar represents 20 µm. (E) GFP (red) shows the *Lfng* expression and NICD (green) identifies the cells with NOTCH1 activity. Scale bar represents 20 µm. (F) Representative bright field and co-IF images of *Lfng* KD and control organoids indicating *Lgr5*-GFP (green) expression. MUC2 (red) marks goblet cells. DAPI (blue) labels nuclei and scale bar represents 25 µm. (G) Representative plots indicating gated percentage of *Lgr5*+ (GFP<sup>high</sup>) and goblet cells (UEA-1<sup>+</sup>/CD24<sup>-</sup>) of *Lfng* KD and control organoids. (H) Ratio of goblet cells to *Lgr5*-GFP + CBCs as determined by flow cytometry. The experiment was performed in triplicate and presented mean ± s.d. (***p<0.001).
DOI: https://doi.org/10.7554/eLife.35710.011
The following figure supplement is available for figure 4:

**Figure supplement 1.** Characterisation of *Lfng* KD organoids.
DOI: https://doi.org/10.7554/eLife.35710.012

quantified in villus crypt units (VCU) of the small intestine (*Ishikawa et al., 1997*). Immunofluorescence analysis based on MUC2 expression in small intestinal tissues from $Lfng^{-/-}$ mice showed an increase in the number of goblet cells when compared to the control ($Lfng^{+/+}$) mice (*Figure 5B*). Finally, using flow cytometry we quantified goblet cell numbers in $Lfng^{-/-}$ mice: 14.1% of small intestinal cells, which is significantly higher than the 7.9% goblet cells in the small intestine of wild-type litter-mate control mice (*Figure 5C*). We observed no change in the Paneth cell numbers after loss of *Lfng* (*Figure 5—figure supplement 1C*).

## *Lfng* deletion reduces Notch signalling

Suppression of Notch signalling is known to increase the goblet cell numbers (*van Es et al., 2005*). We isolated and analysed intestinal progenitor cells (CD24<sup>lo</sup>CD44<sup>+</sup>CD166<sup>+</sup>GRP78<sup>+</sup>) from $Lfng^{+/+}$ and $Lfng^{-/-}$ mice using an established protocol (*Wang et al., 2013*) (*Figure 5D*). RT-qPCR measurements indicated decreased *Hes1* and increased *Atoh1* (transcriptional factor essential for generating secretory lineage (*Shroyer et al., 2007*)) expression in $Lfng^{-/-}$ progenitors compared to the control. We also confirmed reduced activated Notch1 (NICD) in the upper crypts of *Lfng* null mice intestines (*Figure 5—figure supplement 2A,B*). Therefore, *Lfng* silencing appears to lower Notch activity in the progenitors and promotes the secretory lineage leading to an increase in goblet cell numbers (*Zheng et al., 2011*; *Kim and Shivdasani, 2011*).

## Secreted LFNG plays no apparent function

Previous reports have indicated that *Lfng* may be secreted into the extracellular space (*Shifley and Cole, 2008*; *Williams et al., 2016* ). We first examined the medium from intestinal organoid cultures derived from *Lgr5*-GFP mice using solid-phase ELISA. Secreted *Lfng* was detected at a concentration of approximately 315–325 ng/mL using two independent LFNG primary antibodies (*Figure 6A* and *Figure 6—figure supplement 1A*). The other Fringes, RFNG and MFNG, were not detected in the culture medium (*Figure 6—figure supplement 1B,C*). We tried to understand if secreted LFNG influences Notch signalling by affecting receptors in a non-cell autonomous manner. As before, single *Lgr5*-GFP + CBCs were transduced with *Lfng* shRNA and propagated as organoids followed by incubation with conditioned medium harvested from wild-type organoids that contained soluble form of secreted LFNG (sLFNG). After 24 hr, organoid cultures were analysed using flow cytometry, which showed that the percentage of goblet cells remained similar to the *Lfng* knockdown (shRNA) condition and significantly higher than scrambled shRNA-expressing organoids which express endogenous LFNG (*Figure 6B*). RT-qPCR revealed that the expression levels of Notch ligands DLL1 and DLL4 were similar between *Lfng* knockdown with and without soluble LFNG (*Figure 6—figure supplement 1D*).

We then examined intestinal tissues from mutant *Lfng* mice ($Lfng^{RLFNG/+}$ or *RLfng*) in which the N-terminal sequence of LFNG, which normally allows for protein processing and secretion, is replaced with the N-terminus of Radical fringe (a Golgi resident protein) (*Williams et al., 2016*) (*Figure 6—figure supplement 1E*). IF analysis based on MUC2 expression in small intestinal tissues from *RLfng* mice showed similar goblet cell numbers in villus crypt units compared to wild-type ($Lfng^{+/+}$) mice (*Figure 6C*). Taken together, our in vitro and in vivo findings suggest that the effect of LFNG on goblet cell numbers and intestinal homeostasis is not dependent on its secreted form.

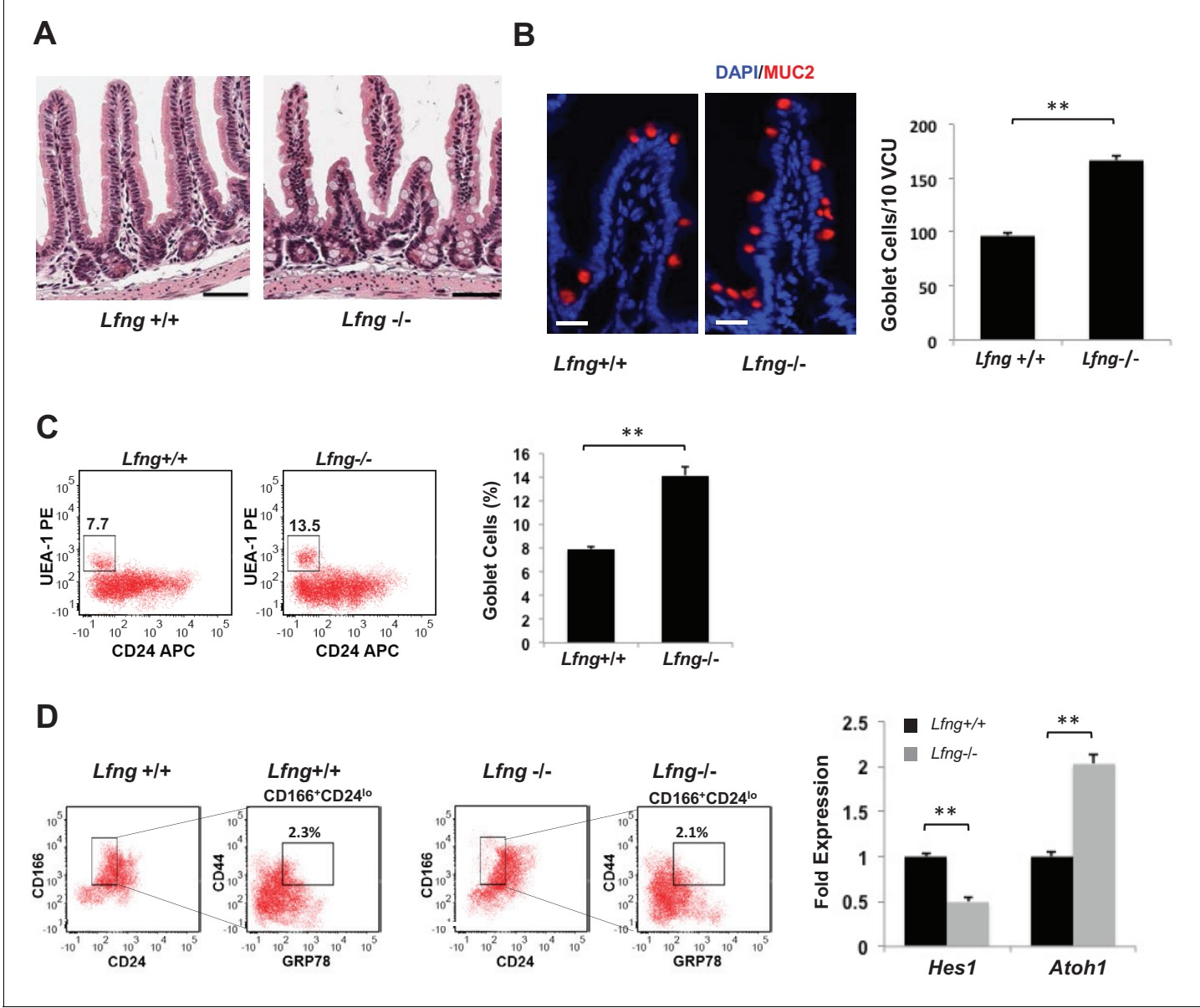

**Figure 5.** *Lfng* loss results in increased goblet cell differentiation in vivo. (**A**) Representative H and E sections from the small intestine of *Lfng*[+/+] and *Lfng*[−/−] mice. Scale bar represents 50 μm. (**B**) Left: Representative IF images of intestine of *Lfng*[+/+] and *Lfng*[−/−] mice. MUC2 (red) marks goblet cells. DAPI (blue) labels nuclei. Right: Quantification of the number of goblet cells of n = 4 mice/condition and n = 500 VCU per mouse presented as mean ± s.d. (**C**) Left: Representative plots indicating gated percentage of goblet cells (UEA-1[+]/CD24[-]) from intestinal tissue derived from *Lfng*[+/+] or *Lfng*[−/−] mice. Right: Percentage of goblet cells presented as mean ± s.d. The data represent n = 3 mice/condition. (**D**) Left: Representative plots indicating gated population of intestinal progenitors from the intestine of *Lfng*[+/+] and *Lfng*[−/−] mice. Percentage reflects fraction of total population. Right: RT-qPCR measurements in progenitor cells from *Lfng*[+/+] and *Lfng*[−/−] mice. The experiment was performed in triplicate presented as mean ± s.d. (**p<0.01).

DOI: https://doi.org/10.7554/eLife.35710.013

The following figure supplements are available for figure 5:

**Figure supplement 1.** Histological and flow cytometric analysis of *Lfng* null intestines.
DOI: https://doi.org/10.7554/eLife.35710.014

**Figure supplement 2.** *Lfng* loss results in reduced Notch activity.
DOI: https://doi.org/10.7554/eLife.35710.015

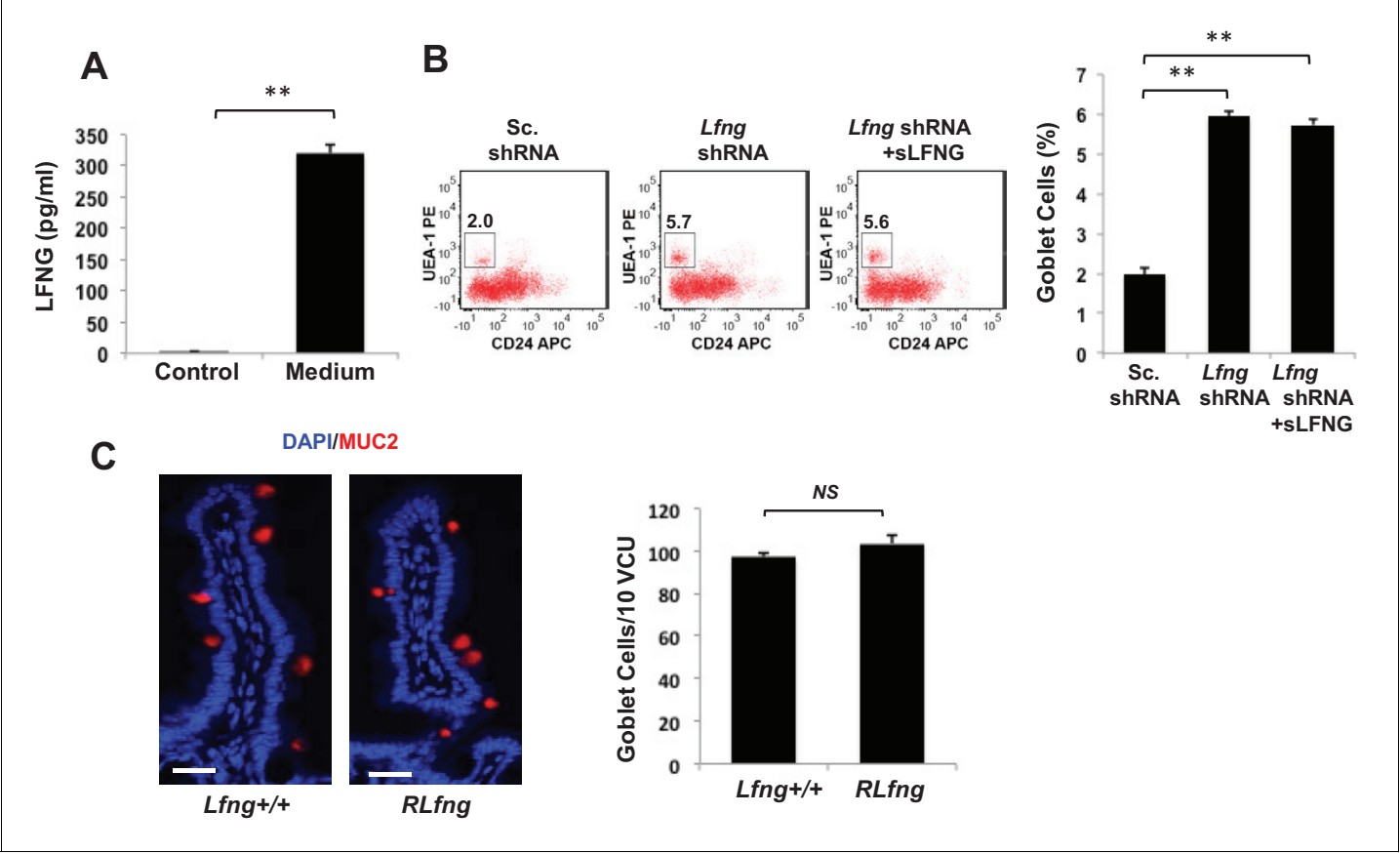

**Figure 6.** Secreted LFNG plays no apparent function. (**A**) ELISA of the secretion of LFNG in culture medium from *Lgr5*-GFP organoids. Culture medium (T = 0 days) was used as a control. The experiment was performed in triplicate and presented as mean ± s.d. (**B**) Left: Representative plots indicating gated percentage of goblet cells (UEA-1$^+$/CD24$^-$) for organoids under Sc. shRNA control, *Lfng* KD and *Lfng* KD treated with sLFNG conditions. Right: Percentage of goblet cells in each condition. The experiment was performed in triplicate and presented as mean ± s.d. (**C**) Left: Representative IF images of intestine of *Lfng*$^{+/+}$ and *Lfng*$^{RLfng/+}$ mice. MUC2 (red) marks goblet cells. DAPI (blue) labels nuclei. Right: Quantification of the number of goblet cells of n = 4 mice/condition and n = 500 VCU/mouse. Data presented as mean ± s.d. (**p<0.01).
DOI: https://doi.org/10.7554/eLife.35710.016

The following figure supplement is available for figure 6:

**Figure supplement 1.** Secretion of Fringe proteins.
DOI: https://doi.org/10.7554/eLife.35710.017

## LFNG promotes DLL expression on the cell surface

In order to understand if LFNG, like RFNG, can affect the cell surface expression of DLL, we examined ligand availability and concentration on the cell surface. Seven days old *Lfng* KD and control organoids were dissociated and single unpermeabilised cells were labelled with CD24 and UEA-1 to mark goblet cells and NOTCH1-Fc to quantify ligand binding to NOTCH1 (*Figure 7A*). Mean fluorescent intensity of NOTCH1 binding to goblet cells with *Lfng* knockdown was reduced compared to the control, suggesting that the ligands available on the goblet cell surface for NOTCH1 to bind to were reduced after the loss of *Lfng*. Flow cytometry shows that DLL1 and DLL4 levels, detected using specific antibodies, on the goblet cell surface reduced after the loss of *Lfng* (*Figure 7B*), although the total expression of DLL1 and DLL4 by the goblet cells measured after the permeabilising the cells remained almost the same (*Figure 7C*). Western blotting also confirmed that the total DLL1 and DLL4 expression in goblet cells does not change significantly after the loss of *Lfng* (*Figure 4—figure supplement 1D*).

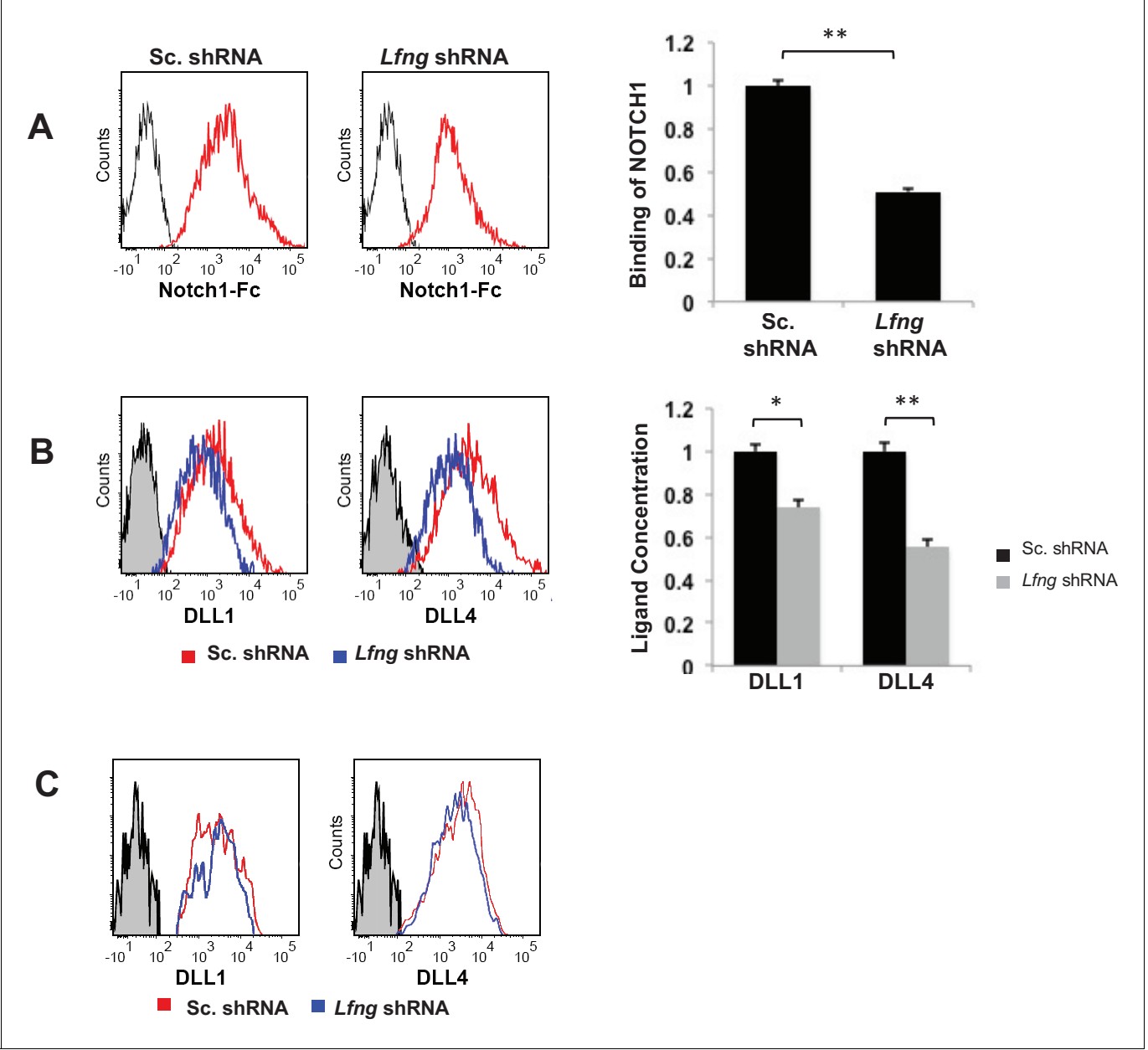

**Figure 7.** LFNG promotes cell surface expression of DLL. (**A**) Ligand availability on *Lfng* KD and Sc. Control goblet cells. Representative traces (left) and MFI (right) showing ligand binding to NOTCH1 measured by flow cytometry. Unstained goblet cells were used as a negative control. The experiment was performed in triplicate and presented as mean ± s.d. (**D**) Cell surface DLL1 and DLL4 concentration on *Lfng* KD and Sc. Control unpermeabilised goblet cells. Left: Representative traces measured by flow cytometry. Right: MFI measurements. The experiment was performed in triplicate and presented as mean ± s.d. (**E**) Cell surface DLL1 and DLL4 concentration on *Lfng* KD and Sc. Control permeabilised goblet cells. (**p<0.01).
DOI: https://doi.org/10.7554/eLife.35710.018

## Discussion

We report that *Rfng* is enriched in the Paneth cells and promotes cell surface expression of DLL1 and DLL4. This promotes Notch activity in the neighbouring *Lgr5+* CBCs assisting their self-renewal. *Mfng* does not appear to contribute significantly in maintaining the epithelium. *Lfng* on the other hand is expressed by enteroendocrine, Tuft and goblet cells and suppresses the secretory lineage (*Figure 8*). Even though Fringe proteins do not appear to be essential, they provide another layer of spatial and lineage-specific modulation that might enhance robustness of intestinal homeostasis.

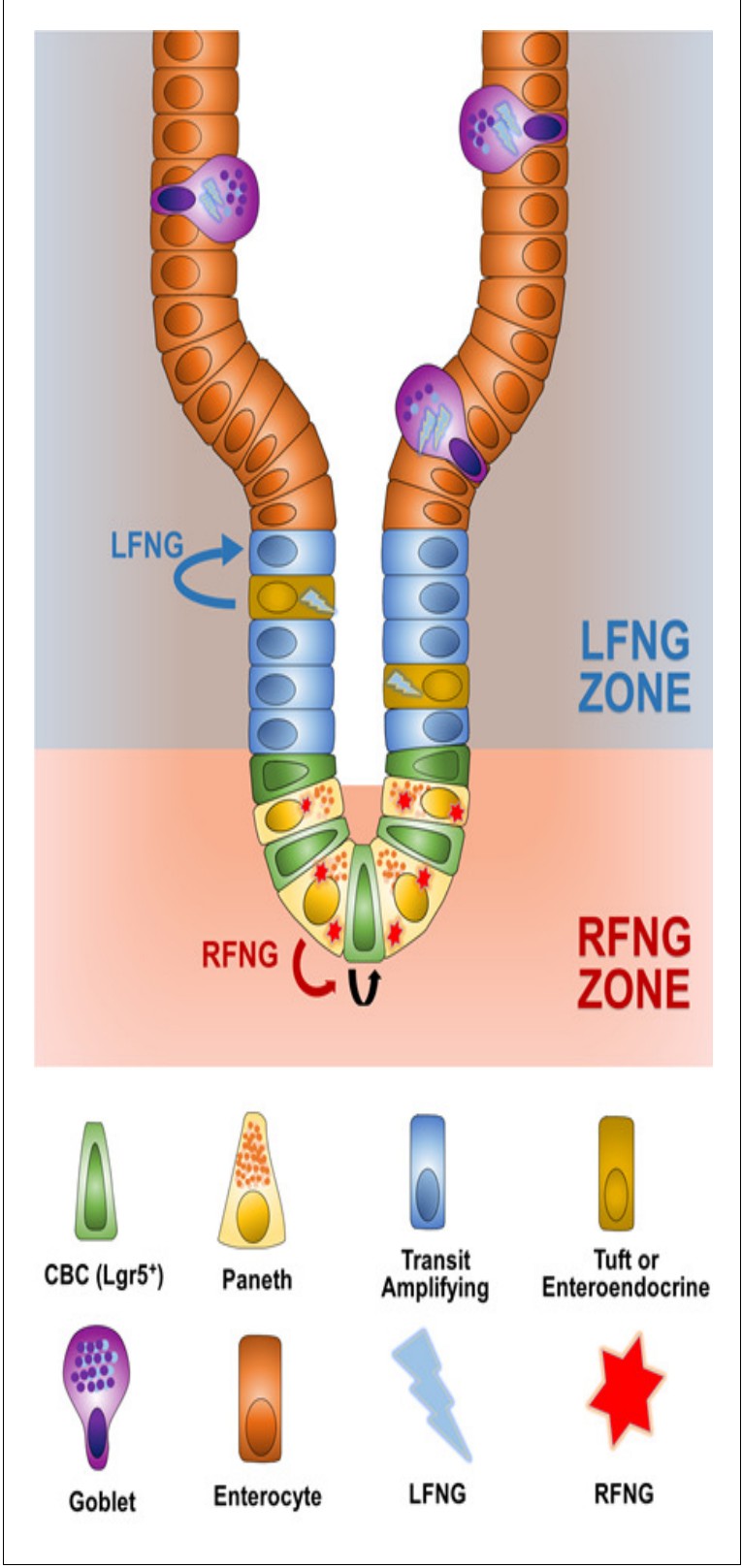

**Figure 8.** Summary. *Rfng* is enriched in the Paneth cells and promotes cell surface expression of DLL1 and DLL4. This promotes Notch activity in the neighbouring *Lgr5+* CBCs assisting their self-renewal. *Mfng* does not appear to contribute significantly in maintaining the epithelium. *Lfng* on the other hand is expressed by enteroendocrine, Tuft, and goblet cells and suppresses the secretory lineage.

*Figure 8 continued on next page*

*Figure 8 continued*

DOI: https://doi.org/10.7554/eLife.35710.019

This is consistent with the highly robust regulation of Notch activity in the intestinal epithelium as inhibition of *Notch1* or *Dll1* only causes minor defective phenotype, while inhibition of *Notch2*, *Dll4*, *Jag1*, *Hes1*, *Hes3* or *Hes5* causes no significant phenotype (*Pellegrinet et al., 2011*; *Wu et al., 2010*; *Ueo et al., 2012*).

We have observed that both RFNG and LFNG can increase the presence of DLL1 and DLL4 on the plasma membrane. This can potentially contribute to the increase in cis-inhibition of NOTCH1 by DLL1 in the presence of fringe (*LeBon et al., 2014*). Fringe modulation of ligands will be of significance in understanding Notch activity in cancer stem cell asymmetric division where LFNG, DLL1 and NOTCH1 are present in the same cell (*Bu et al., 2013*; *Bu et al., 2016*). However, the mechanism behind the increase in cell-surface expression of the ligands still needs to be understood. The glycosylation state of proteins has been known to affect their intracellular trafficking (*Huet et al., 2003*; *Ohtsubo and Marth, 2006*). It raises the possibility that fringe mediated glycosylation or the addition of Galactose and Sialic acid post fringe activity might affect the trafficking of DLLs to the cell surface. *Lfng* in the *Dll1* expressing cell, in the presence of *Dll3*, is known to reduce Notch activity in the neighbouring cell (*Okubo et al., 2012*). This raises the possibility that ligands interact with each other in the presence of *Lfng* which might explain our observation. *In vitro* reductionist studies may need to be conducted in systems expressing single ligand and fringe to understand the mechanism in detail. Also, our experiments cannot completely rule out the possibility that low levels of *Rfng* expression in CBCs (in comparison to Paneth cells) can also contribute to the phenotype by directly modulating Notch receptors. We have observed some mesenchymal cells also express *Rfng* detectable by RNA ISH. We also observe that some of the mesenchymal cells also express *Dll1* (*Figure 2—figure supplement 1*). Further studies are necessary to map the expression of all the Notch ligands in different mesenchymal cell types. This raises the possibility that the mesenchyme can also provide Notch ligands to the *Lgr5* + CBCs *in vivo*. In case that is true, our proposed mechanism that *Rfng* promotes cell surface expression of *Dll1* might be applicable to the mesenchymal cells too. Upon loss *Rfng*, reduced *Dll1* expression on the cell surface of Paneth cells and mesenchymal cells would result in reduced Notch activity in the CBCs, as observed. However, the crypt cells are separated from the mesenchyme by the basement membrane. The efficacy of DLL mediated Notch signalling across the intestinal basement membrane needs to be explored.

Although we have observed that the *Lfng* expressing cells are found both in the upper crypt and in the villus, our data suggests that LFNG in NICD- post-mitotic secretory cells of the upper crypt promotes Notch activity in the neighbouring enterocyte progenitors. As Notch signalling is not active in the villus, the *Lfng*+ cells of the villi likely do not impact epithelial cell differentiation. It would be interesting to explore the reason secretory cells expressing Notch ligands and *Lfng* are found in the villi. The differences, other than functional consequence, between the *Lfng*+ cells of the upper crypt and villus needs to be explored.

Notch pathway is a potential therapeutic target, but blocking the pathway leads to serious GI related side effects (*van Es et al., 2005*). Targeting the Notch pathway through fringe appears to be a potentially viable strategy to exclusively modulate intestinal epithelial regeneration or its functions, absorption and mucus secretion, as Notch activity in the stem cells or progenitors can be specifically targeted by blocking *Rfng* or *Lfng* respectively.

## Materials and methods

### Mice

*Lgr5*-GFP (Jackson Lab #8875, RRID:IMSR_JAX:008875) strain has been described previously (*Sato et al., 2009*). *Lfng* null (*Lfng*$^{tm1Rjo}$), *Mfng* null (*Mfng*$^{tm1Seco}$, RRID:MGI:3849430) and *Rfng* null (*Rfng*$^{tm1Tfv}$) mice were maintained as described here (*Moran et al., 2009*; *Ryan et al., 2008*). *Lfng*$^{RLFNG/+}$ mice were maintained as previously described (*Williams et al., 2016*). Littermates of Fringe mutants with wild-type gene expression were used as controls. *Lfng*-GFP (GENSAT # RRID: MMRRC_015881-UCD) were received as FVB/N - C57BL/6 hybrids and crossed to C57BL/6 mice for

at least 10 generations (*Gong et al., 2003*; *Semerci et al., 2017*). All procedures were conducted under protocols approved by the appropriate Institutional Animal Care and Use Committees at Ohio State University (# 2012A00000036-R1), Duke University (# A286-15-11), Baylor College of Medicine (# AN-5004), Cornell University (# 2010–0100) or Research Animal Resource Center of Weill Cornell Medical College (# 2009–0029).

## Organoid culture and flow cytometry

Organoids from *Lgr5*-GFP mouse intestines were cultured as previously described with minor modifications (*Sato et al., 2011*; *Sato et al., 2009*). Briefly, small intestines were harvested, washed with PBS and opened up longitudinally to expose luminal surface. A glass coverslip was then gently applied to scrape off villi and the tissue was cut into 2–3 mm fragments and incubated with 2 mM EDTA for 45 min on a rocking platform at 4°C. EDTA solution was then decanted and replaced with cold PBS. The tissues were vigorously agitated to release the crypts. Next, single cell dissociation was achieved by incubating purified crypts at 37°C with Trypsin-EDTA solution containing 0.8KU/ml DNase, 10 μM Y-27632 for 30 min. To isolate *Lgr5*-GFP+ cells, single cells were resuspended in cold PBS with 0.5% BSA and GFP$^{high}$ cells were sorted by FACS (Beckman Coulter/BD FACSAria).

Dissociated cells were also stained with anti-CD24 antibody and UEA-1. Paneth cells were sorted based on side scatter and CD24 expression (CD24$^{high}$/SSC$^{high}$) and goblet cells were identified as UEA-1$^{+}$/CD24$^{-}$ (*Sato et al., 2011*; *Wong et al., 2012*). Viable cells were gated based on 7-AAD or Sytox blue staining. Data analysis was performed using FlowJo software.

Single *Lgr5*-GFP+ CBCs were plated in Matrigel and cultured in medium containing: Advanced DMEM/F12 supplemented with Glutamax, 10 mM HEPES, N2, B27 without vitamin A, 1 μM N-acetyl-cysteine, 50 ng/mL EGF, 100 ng/mL Noggin, and 10% R-SPONDIN1 conditioned medium.

Lentiviral constructs containing *Lfng* shRNA (sc-39491-SH), *Mfng* shRNA (sc-39493-SH), *Rfng* shRNA (sc-39495-SH), or scrambled shRNA (sc-108060) were purchased from Santa Cruz Biotechnology. Lentiviral transduction of *Lgr5*-GFP CBCs were performed by 'spinoculation' method described previously (*Koo et al., 2011*). Transduced CBCs were cultured as organoids and analysed after 7 days. RBPJκ-dsRed reporter (Addgene #47683) was transfected into single Sc. shRNA-expressing or *Rfng* shRNA-expressing sorted *Lgr5*-GFP CBCs using Lipofectamine-2000 as described earlier (*Schwank et al., 2013*).

Organoid Reconstitution Assay was performed as described previously (*Rodríguez-Colman et al., 2017*). Briefly, FACS sorted Paneth cells and *Lgr5*-GFP+ CBCs were mixed, spun down and incubated at room temperature for 10 min. The pellet was then plated in Matrigel.

## RT-qPCR and protein analysis

A Qiagen RNeasy kit was used to extract total RNA. RT-PCR primers from Genecopoeia were used for the following genes: β-*Actin*, *Lgr5*, *Lfng*, *Mfng* and *Rfng*. Taqman primers (ABI) were used for: *Lgr5*, *Notch1*, *Hes1*, *Hes5*, *Dll1*, *Dll4*, and *Jag1*. *Gapdh* or β-*Actin* was used as internal control. Protein isolation and western blotting were performed as previously described, using β-ACTIN for normalisation (*Pan et al., 2008*). ELISA kits for LNFG, RFNG, and MFNG were purchased from MyBioSource and assays were performed according to the manufacturer's instructions similar to the following protocol. Solid-phase ELISA assays were independently conducted using LFNG, RFNG, and MFNG antibodies (referred to as antibody-2) purchased from Santa Cruz Biotechnology for verification of results obtained from the corresponding kits.

## Ligand availability assay

Ligand availability assays were performed as previously described (*LeBon et al., 2014*). Briefly, blocking buffer (PBS, 2% FBS, 100 μg/mL CaCl$_2$) and binding Buffer (PBS, 2% BSA, 100 μg/mL CaCl$_2$) were prepared. Subsequently, cells were incubated in blocking buffer for 30 min at 37°C followed by incubation with 0.5 μg/mL NOTCH1-Fc (R and D #5267) diluted in binding buffer for 1 hr at 4°C. Cells were then washed three times in blocking buffer and incubated in secondary antibody diluted in binding buffer for 40 min at room temperature. Finally, cells were washed three times in blocking buffer and analysed by flow cytometry.

## Immunofluorescence (IF) and immunohistochemistry (IHC)

Sections of paraffin embedded tissues were deparaffinised using Xylene and rehydrated. Antigen retrieval was performed using Proteinase K (Dako) or 10 mM Tris buffer at pH9. The sections were incubated in Protein Block (Dako) for 10 min at room temperature (RT). Primary antibodies diluted in Antibody Diluent (Dako) were added and incubated overnight at 4°C. Slides were then washed in PBS and incubated in secondary antibodies diluted in Antibody Diluent for 1 hr at RT and washed in PBS. The slides were then mounted using Vectashield mounting medium containing DAPI. Intestinal sections were stained with Haematoxylin and Eosin (H and E), Periodic Acid-Schiff (PAS), Alcian Blue (AB) or Nuclear Fast Red according to standard methods. Intestinal organoids embedded in Matrigel were fixed with 3% PFA for 15 min at room temperature and permeabilised using 0.2% Triton-X for IF according to the protocol described above. Antibodies used are listed in supplementary methods. Antibodies used are listed in *Supplementary file 1*.

Protocol was modified while staining for Notch1 intracellular domain (NICD). Antigen retrieval was performed using Trilogy (Cell Marque). Sections were then incubated in 3% hydrogen peroxide diluted in PBS for 10 min. Protein blocking, primary antibody and secondary antibody incubation were performed as described above. Signal was further amplified using TSA Plus Fluorescein kit (Perkin Elmer). To quantify, NICD+ nuclei and total number of nuclei (based on DAPI signal) were counted in each crypt (35 to 50 crypts from each section) to obtain the fraction of NICD+ nuclei.

0.5 mg EdU in 150 µl PBS (~16.66 mg/kg) was injected intraperitoneally into *Lfng*-GFP mice two hours prior to euthanasia (*Kabiri et al., 2014*). Incorporated EdU was detected using Click-It EdU imaging kit (Thermo Fisher #C10640).

## RNA in situ hybridisation (ISH)

RNA-ISH was performed using RNAscope 2.5HD duplex assay kit (ACDBio) (*Wang et al., 2012*) as per manufacturer's instructions. Briefly, the assay was first validated by using a positive control probe for *Polr2a* and a negative control probe for a bacterial gene *dapB*. Probes, labelled with Alkaline Phosphatase, for *Rfng* or *Dll1* were hybridised to the tissue sections. The signal was generated using Fast Red substrate. The slides were washed in water and then PBS and were stained for Lysozyme protein as described above in the Immunofluorescence section. Fast Red signal was detected by a fluorescence microscope as described in (*Lauter et al., 2011*).

## Statistical analysis

The data is represented as mean ± S.E.M (standard error of mean) unless otherwise indicated. A Student t-test was applied to compare two groups using $p<0.05$ to establish statistical significance.

# Acknowledgement

We are grateful to Dr Brigid Hogan of Duke University, Dr Linda Samuelson of University of Michigan and Dr Robert Haltiwanger of University of Georgia for their valuable comments. We thank Dr Steven Lipkin of Weill Cornell Medical College for providing R-SPONDIN1 conditioned medium. We thank Dr Kameswaran Surendran of Sanford Research for sharing the protocol for NICD staining.

# Additional information

## Funding

| Funder | Grant reference number | Author |
| --- | --- | --- |
| National Institutes of Health | R35GM122465, R01GM114254 | Xiling Shen |
| Defense Advanced Research Projects Agency | HR0011-16-C-0138 | Xiling Shen |
| National Science Foundation | NSF 1350659 career award | Xiling Shen |

The funders had no role in study design, data collection and interpretation, or the decision to submit the work for publication.

## Author contributions
Preetish Kadur Lakshminarasimha Murthy, Conceptualization, Data curation, Formal analysis, Investigation, Visualization, Writing—original draft, Project administration, Writing—review and editing; Tara Srinivasan, Data curation, Formal analysis, Investigation, Visualization, Writing—original draft, Writing—review and editing; Matthew S Bochter, Resources, Maintained the mouse colonies and, harvested and processed intestinal samples; Rui Xi, Formal analysis, Investigation, Visualization; Anastasia Kristine Varanko, Kuei-Ling Tung, Investigation, Performed experiments to generate supporting data not shown in the manuscript; Fatih Semerci, Keli Xu, Mirjana Maletic-Savatic, Resources, Writing—review and editing; Susan E Cole, Conceptualization, Resources, Writing—review and editing; Xiling Shen, Conceptualization, Supervision, Funding acquisition, Writing—review and editing

## Author ORCIDs
Preetish Kadur Lakshminarasimha Murthy (ID) http://orcid.org/0000-0002-9762-8376
Matthew S Bochter (ID) http://orcid.org/0000-0001-9607-3770
Fatih Semerci (ID) http://orcid.org/0000-0002-0512-1827
Mirjana Maletic-Savatic (ID) http://orcid.org/0000-0002-6548-4662
Xiling Shen (ID) http://orcid.org/0000-0002-4978-3531

## Ethics
Animal experimentation: All procedures were conducted under protocols approved by the appropriate Institutional Animal Care and Use Committees at Ohio State University (# 2012A00000036-R1), Duke University (# A286-15-11), Baylor College of Medicine (# AN-5004), Cornell University (# 2010-0100) or Research Animal Resource Center of Weill Cornell Medical College (# 2009-0029).

## Decision letter and Author response
Decision letter https://doi.org/10.7554/eLife.35710.029
Author response https://doi.org/10.7554/eLife.35710.030

# Additional files

## Supplementary files
• Supplementary file 1. Supplementary Methods. List of antibodies used for Immunofluorescence and Western blotting.
DOI: https://doi.org/10.7554/eLife.35710.020
• Transparent reporting form
DOI: https://doi.org/10.7554/eLife.35710.021

## Major datasets
The following previously published dataset was used:

| Author(s) | Year | Dataset title | Dataset URL | Database, license, and accessibility information |
| --- | --- | --- | --- | --- |
| Sato T, van Es JH, Snippert HJ, Stange DE, Vries RG, van den Born M, Barker N, Shroyer NF, van de Wetering M, Clevers H | 2011 | Paneth cells constitute the niche for Lgr5 stem cells in intestinal crypts | https://www.ncbi.nlm.nih.gov/geo/query/acc.cgi?acc=GSE25109 | Publicly available at the NCBI Gene Expression Omnibus (accession no: GSE25109) |

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
