## [Decision Letter]

Thank you for submitting your article "Spatially Specific Fringe Modulation of Notch Ligands Supports Intestinal Homeostasis" for consideration by *eLife*. Your article has been reviewed by three peer reviewers, and the evaluation has been overseen by Fiona Watt as the Senior and Reviewing Editor. The following individuals involved in review of your submission have agreed to reveal their identity: Owen Sansom (Reviewer #2); Kevin Myant (Reviewer #3).

The reviewers have discussed the reviews with one another and the Reviewing Editor has drafted this decision to help you prepare a revised submission.

Summary:

This manuscript describes interesting studies on the expression and possible function of Fringe proteins within the intestinal epithelium. The authors make the argument that several Fringe proteins are expressed highly in Paneth cells, where they increase the cell surface expression of DLL1 and DLL4, two Notch ligands. In this manner, they are proposed to regulate Notch signalling in *Lgr5*+ intestinal stem cells.

Essential revisions:

None of the reviewers was completely convinced that the authors have shown what they claim and further experiments and/or analysis are needed for the work to be convincing. In particular the authors need to more clearly demonstrate that Fringe proteins are expressed in Paneth cells and that their cell sorting technique is robustly sorting Paneth / *Lgr5*+ cells.

1) Expression of Fringe proteins in Paneth cells. Although this is a key issue for the study, the data is limited and needs to be strengthened. While Fringe proteins were shown by Schroder et al. to be expressed in the intestinal crypts, they did not appear to be particularly specific in their in situ studies for the location of Paneth cells. No additional localization by either ISH or IHC is shown. While the FACS studies "using an established protocol" suggest greater expression in Paneth cells compared to *Lgr5*+ cells, the methodology is not easy to decipher, and the relative specificity of the FACS studies is not convincing. The authors need to describe exactly which surface markers were used to achieve their claimed populations. Validation of the FACS studies is needed, together with immunolocalization studies.

2) Expression of DLL1 and DLL4 on Paneth cells. The authors suggest that the ligands DLL1 and DLL4 are well expressed on Paneth cells, based on Western blotting of "Paneth cells". These are presumably from cell sorting, although the figure legend describes the lysates as derived from "organoids". Stamataki D et al. (PLoS One 2011) used a DLL1-β-galactosidase transgene to mark DLL1+ cells and found it was expressed in goblet cells (96%) and enteroendocrine cells (66%) but not in Paneth cells. Van et al., 2012 used a different DLL1-GFP-IRES-CreERT2 knockin mouse; strong expression above the stem cell zone (e.g. position +5 – +10) and only weak expression in the Paneth cells was reported. Finally, Shimizu H et al. (PeerJ, 2014) used double immune staining to show that DLL1 was not expressed in Paneth cells. In contrast Shimizu et al. did confirm that DLL4 was expressed in Paneth cells. Given these published findings, the authors need substantial additional evidence (e.g. immunofluorescence, qPCR, in situ hybridisation) in order to claim DLL1 expression in Paneth cells.

3) The conclusion that *Rfng* expression on Paneth cells promotes Notch activity in neighbouring *Lgr5*+ CBCs is based on the finding that *Rfng*^-/-^ intestines show reduced CBCs, although no other phenotype was detected, which is inconsistent with a reduced Notch phenotype. In addition, loss of *Rfng* led to a reduction in *Lgr5*+ CBCs in organoid culture. However, this conclusion that the changes are due to "Paneth cells" remains puzzling, given the reported absence of DLL1 in Paneth cells. In addition, several groups (Garabedian EM et al., J Biol Chem 1997; Kim TH et al., Proc Natl Acad Sci U S A 2012) have now shown that deletion in vivo of Paneth cells does not result in any significant in vivo phenotype. While Paneth cells likely express *Dll4*, depletion of *Dll4* alone or administration of a *Dll4*-immunoneutralizing antibody has no effect on intestinal epithelial homeostasis (Ridgway et al., 2006). This would be consistent with the lack of a major role for Paneth cells in regulating the ISC compartment. To support the conclusion that Paneth cell numbers are reduced after *Rfng* deletion in vivo Figure 1, qPCR for LYZ needs to be done on material from Figure 1F. Also *Lgr5* expression analysis would be better than OLFM4 and ASCL2 in terms of consistency. In all the authors' experiments CBCs are also depleted for RFNG (the organoids are derived from *Rfng* knockdown CBCs and the mice are whole body knockouts). Therefore, it is impossible to rule out a role for RFNG expressed in CBCs or other cell types. To address this the authors should directly mix *Rfng* knockdown Paneth cells with wildtype CBCs (and vice-versa) and determine whether colony forming capacity is impacted.

Other comments:

1) The authors state that "Notch signalling promotes self-renewal of CBCs" and reference the review by van der Flier and Clevers. In that review, inhibition of Notch leads to goblet differentiation, reduced proliferation, and fewer CBCs; however, this is often reversible. The authors should revise their statements accordingly.

2) The lysozyme expression appears very nuclear in Figure 1C. Conventional immunohistochemistry is required to assess the different cell populations clearly. The same nuclear localisation for the *Lgr5*-GFP is not in line with the previous published expression patterns and should be re-evaluated by the authors.

3) Given that goblet cell numbers vary along the length of the small intestine (SI) it is important the same parts of the small intestine are examined e.g. In some of the figures it appears that different parts of the SI have been analysed. The authors need to state which part has been analysed in their scoring and if not the same parts must be analysed.

4) To clearly show a reduction in Notch signalling activity after *Lfng* loss, the authors should examine the expression of Notch intra cellular domain (Nicd).

5) Given that Notch signalling is not active in the villus, the function of *Lfng* in goblet cells on Notch signalling within these cells is confusing. A more likely explanation is that *Lfng* positive progenitors contribute to the described effects on colony formation as well as differentiation. The authors should clarify whether this progenitor population is controlling the detected effects rather the expression in goblet cells. For example, by *Lfng*-GFP sorting and exclusion of *Muc2* positive cells, or *Lfng*-GFP and CD44 double positive cell sorting, followed by functional assays.

[Editors’ note: this article was subsequently rejected after discussions between the reviewers, but the authors submitted a revised version at a later point, being the manuscript then accepted for publication.]

Thank you for submitting your work entitled "Spatially Specific Fringe Modulation of Notch Ligands Supports Mammalian Intestinal Homeostasis" for consideration by *eLife*. Your article was sent to the three peer reviewers of the previous version of the manuscript, and the evaluation has been overseen by a Senior/Reviewing Editor. The following individuals involved in review of your submission have agreed to reveal their identity: Kevin Myant (Reviewer #3). Reviewer #1 did not agree to participate in the review process and so the decision regarding the manuscript was based on discussions between Reviewers #2 and #3 and the Senior Editor.

As you can from the verbatim comments, reviewer #3 was satisfied with your revisions. However, reviewer #2 felt that you had failed to address key points raised previously. In consultation, the Senior Editor agreed with reviewer #2 that your revisions were inadequate. Therefore we have agreed that the manuscript cannot be published in *eLife*.

*Reviewer #2:*

The authors have attempted to revise the manuscript in response to my comments. Unfortunately the response to my comments has not really allayed my problems with the manuscript.

The four points below have not, in my mind, been satisfactorily addressed and need to be.

Point 1: Expression of fringe.

The ISH data are not convincing for many reasons. First, RNAscope normally doesn't give dominantly signal in the nucleoli, based on the fact that mRNA is shuttled out of the nucleus after synthesis quickly. Additionally, why are Lysozyme positive cells in picture Figure 1B are not positive for *Rfng*? Quantification of *Rnfg* positive Paneth cells will be useful to understand the importance of *Rnfg* in Paneth cells. Second, cells outside the crypt are positive for *Rnfg*. How do the authors interpret these *Rnfg*-positive cells? And how do these cells contribute to the detected effects?

Point 2: Lysozyme expression.

The enlarged figure is not sufficient and doesn't show cytoplasmic lysozyme expression at all. This technical issue needs to be corrected and much more convincing data needs to be provided. Data shown in Figure 1B show expression patterns for Lysosyme as expected and indicates that correct staining is possible.

Point 4. To clearly show a reduction in Notch signalling activity after *Lfng* loss, the authors should examine the expression of Notch intra cellular domain (Nicd).

It appears that the intensity of the IF presented in the revised version of the manuscript of NICD is much stronger rather than a change in NICD positive cells. The authors need to quantify this data for both points to be able to show the effect of *Lfng* loss. Furthermore, double staining of *Lfng*-GFP and NICD will help to understand the interaction of *Lfng* positive and Notch positive cells.

Point 5. Given that Notch signalling is not active in the villus, the function of *Lfng* in goblet cells on Notch signalling within these cells is confusing.

Data which cell type is actually *Lfng* positive is not solid. Figure 4 doesn't show convincingly that *Lfng* is expressed in transit amplifying cells. This is of major importance since this would be the compartment that shows overlap with Notch activity and would explain the *Lfng* loss mediated effect on Notch signalling. *Lfng* positive Goblet cells in the villus can't explain this effect.

*Reviewer #3:*

My main concerns have been suitably addressed and I would now recommend publication of this study.

---

## [Author Response]

Essential revisions:None of the reviewers was completely convinced that the authors have shown what they claim and further experiments and/or analysis are needed for the work to be convincing. In particular the authors need to more clearly demonstrate that Fringe proteins are expressed in Paneth cells and that their cell sorting technique is robustly sorting Paneth / Lgr5+ cells.1) Expression of Fringe proteins in Paneth cells. Although this is a key issue for the study, the data is limited and needs to be strengthened. While Fringe proteins were shown by Schroder et al. to be expressed in the intestinal crypts, they did not appear to be particularly specific in their in situ studies for the location of Paneth cells. No additional localization by either ISH or IHC is shown. While the FACS studies "using an established protocol" suggest greater expression in Paneth cells compared to Lgr5+ cells, the methodology is not easy to decipher, and the relative specificity of the FACS studies is not convincing. The authors need to describe exactly which surface markers were used to achieve their claimed populations. Validation of the FACS studies is needed, together with immunolocalization studies.

We agree with the reviewers on the need to strengthen the pivotal datum of the study. Firstly, we validated the FACS protocol used to obtain the *Rfng* expression pattern. We confirmed that the Paneth cells sorted-based on side scatter and CD24 expression are indeed positive for their marker Lysozyme (Figure 1—figure supplement 1A). *Lgr5*-GFP CBCs were sorted based on GFP signal (Figure 1—figure supplement 1B). We have updated the Materials and methods section to include the FACS protocol.

We have not been successful in immunostaining for RFNG using any of the commercially available antibodies. An antibody reported to be suitable for immunostaining (Ryan et al., 2008) showed non-specific staining on intestinal sections. We therefore performed in situ hybridisation (ISH) using RNAscope probes (ACDBio) to obtain *Rfng* expression pattern and the results of which show that *Rfng* transcripts are primarily localised to the Paneth cells (Figure 1B). Paneth cells were identified by Lysozyme expression. Z-stack images were taken to correctly identify the Paneth cells (Author response image 1). Also, our RNA-ISH assay using RNAscope produces similar images to those published previously (Lafkas D et al., Nature, 2015).

**Author response image 1. respfig1:** 3D projection (from z-stack spanning 9μm) was used to correctly identify the Paneth cells in Figure 1B. *Rfng* transcripts (red) and Lysozyme protein (green) expression can be seen at the bottom of the crypts. Dapi (Blue) labels the nuclei.

2) Expression of DLL1 and DLL4 on Paneth cells. The authors suggest that the ligands DLL1 and DLL4 are well expressed on Paneth cells, based on Western blotting of "Paneth cells". These are presumably from cell sorting, although the figure legend describes the lysates as derived from "organoids". Stamataki D et al. (PLoS One 2011) used a DLL1-β-galactosidase transgene to mark DLL1+ cells and found it was expressed in goblet cells (96%) and enteroendocrine cells (66%) but not in Paneth cells. Van et al., 2012 used a different DLL1-GFP-IRES-CreERT2 knockin mouse; strong expression above the stem cell zone (e.g. position +5 – +10) and only weak expression in the Paneth cells was reported. Finally, Shimizu H et al. (PeerJ, 2014) used double immune staining to show that DLL1 was not expressed in Paneth cells. In contrast Shimizu et al. did confirm that DLL4 was expressed in Paneth cells. Given these published findings, the authors need substantial additional evidence (e.g. immunofluorescence, qPCR, in situ hybridisation) in order to claim DLL1 expression in Paneth cells.

We agree that there is a lack of clarity in literature on the expression of *Dll1* in Paneth cells. Studies based on *Dll1* knock-in mice claim that *Dll1* is expressed by Paneth cells. Samataki D et al. (PLoS One 2011, Figure 1D) show that a subset of Paneth cells express the β-galactosidase reporter driven by *Dll1* promoter. Van et al., 2012 show that Paneth cells express DLL1 but the expression level is not as high as that in the secretory progenitors. Although Shimizu H et al. (PeerJ, 2014) claim that Paneth cells do not express DLL1 based on immunostaining experiments, upon closer observation of their data we can see in Figure 3A of their article that DLL1+ cell is found in the region below the progenitors suggesting that it might be a Paneth cell. Shorning BY et al. (PLoS One 2009) have also performed immunostaining for DLL1 albeit using a different antibody and shown that Paneth cells do express DLL1.

Pellegrinet et al., 2011 showed that *Dll4* loss does not significantly affect the intestinal epithelium. However, they also showed that a combined loss of *Dll1* and *Dll4* leads to a near complete loss of stem cells. If *Dll1* were not to be expressed at the bottom of the crypts as claimed by Shimizu H et al. (PeerJ, 2014), combined loss of *Dll1* and *Dll4* would not have such a severe phenotype.

We have been not been successful in immunostaining for DLL1 using commercially available antibodies. Hence, we performed ISH to obtain *Dll1* expression pattern and find that Paneth cells express *Dll1* (Figure 2—figure supplement 1A). We find that the expression level of *Dll1* is very high in a few progenitors in coherence with the data in Van et al., 2012. We also observe that a subset of Paneth cells show higher expression of *Dll1* transcripts compared to others which explains the data in Samataki D et al. (PLoS One 2011).

Together with RNA-ISH (Figure 2—figure supplement 1A), western blot (Figure 2A), flow cytometry (Figure 2D) and cited previous studies we can conclude that Paneth cells express *Dll1* albeit at a lower level than the *Dll1*+ secretory progenitor cells.

We have modified the legend of Figure 2A to correctly indicate that the DLL1 and DLL4 expression data are based on Western blotting of Paneth cells FACS sorted from organoids. We thank the reviewers for pointing out the issue.

3) The conclusion that Rfng expression on Paneth cells promotes Notch activity in neighbouring Lgr5+ CBCs is based on the finding that Rfng^-/-^ intestines show reduced CBCs, although no other phenotype was detected, which is inconsistent with a reduced Notch phenotype. In addition, loss of Rfng led to a reduction in Lgr5+ CBCs in organoid culture. However, this conclusion that the changes are due to "Paneth cells" remains puzzling, given the reported absence of DLL1 in Paneth cells. In addition, several groups (Garabedian EM et al., J Biol Chem 1997; Kim TH et al., Proc Natl Acad Sci U S A 2012) have now shown that deletion in vivo of Paneth cells does not result in any significant in vivo phenotype. While Paneth cells likely express Dll4, depletion of Dll4 alone or administration of a Dll4-immunoneutralizing antibody has no effect on intestinal epithelial homeostasis (Ridgway et al., 2006). This would be consistent with the lack of a major role for Paneth cells in regulating the ISC compartment. To support the conclusion that Paneth cell numbers are reduced after Rfng deletion in vivo Figure 1, qPCR for LYZ needs to be done on material from Figure 1F. Also Lgr5 expression analysis would be better than OLFM4 and ASCL2 in terms of consistency. In all the authors' experiments CBCs are also depleted for RFNG (the organoids are derived from Rfng knockdown CBCs and the mice are whole body knockouts). Therefore, it is impossible to rule out a role for RFNG expressed in CBCs or other cell types. To address this the authors should directly mix Rfng knockdown Paneth cells with wildtype CBCs (and vice-versa) and determine whether colony forming capacity is impacted.

Notch activity is responsible for the maintenance of *Lgr5*+ CBCs at the crypt bottom and differentiation of progenitors into enterocytes in the upper crypt (Noah TK et al., Annu Rev Physiol, 2013; Pellegrinet et al., 2011; Fre et al., 2005). Reduction of Notch signalling owing to the loss of ligands Dll1/4, receptor Notch1 or due to reduction of γ-Secretase activity (Noah TK et al., Annu Rev Physiol, 2013) affects both the upper and lower crypts. However, loss of *Rfng* reduces Notch signalling only in the lower crypts without affecting progenitor differentiation. This explains why we do not observe any goblet cell metaplasia typically associated with reduced Notch activity in the intestine.

We agree that the importance of Paneth cells in vivohas been contested. *Lgr5*+ CBCs are maintained mainly by Notch and Wnt signalling pathways. Both mesenchymal cells and Paneth cells are sources of Wnt ligands (Kabiri Z et al., Development, 2014; Sato T et al., Nature, 2011; Valenta T et al., Cell Rep, 2016). Paneth cells are known to express Notch ligands DLL1 and DLL4 to activate Notch receptors on the *Lgr5*+ CBCs (Sato and Clevers, Science, 2013; Van et al., 2012). Garbedian EM et al. (J Biol Chem, 1997) ablate mature Paneth cells using transgenic mice in which diphtheria toxin or SV40 T antigen is expressed as directed by Cryptidin-2 gene. They observe that the intestinal functions are not significantly affected. They also observe that the region previously occupied by Paneth cells is now occupied by what appear to be transit-amplifying cells. They have not investigated if Notch/Wnt activity has been affected in the *Lgr5*+ CBCs. They have not investigated if the expression of any stem cell markers have been affected (the study was published prior to the identification of *Lgr5*+ cells). It is possible that the “semi-differentiated” cells that replace the Paneth cells provide the necessary Notch ligands to support the neighbouring CBCs. Kim TH et al. (Proc Natl Acad Sci USA, 2012) and Durand A et al. (Proc Natl Acad Sci USA, 2012) ablate Paneth cells by using mice lacking *Atoh1 (Math1*) gene. It has been shown that the role of Notch activity in the intestinal crypts is to suppress *Atoh1* expression (Kazanjian A et al., Gastroenterology, 2010). In the absence of *Atoh1*, suppression of Notch activity by using γ-Secretase inhibitors has no significant phenotype (Van et al., 2010; Kim TH, J Biol Chem, 2010; Kazanjian A et al., Gastroenterology, 2010). This explains why the loss of Paneth cells upon the loss of *Atoh1* does not significantly affect the *Lgr5+ CBCs*. Sato T et al. (Nature, 2012) have shown that Paneth cell depletion leads to a significant reduction in stem cell numbers. Also, Yilmaz OH et al. (Nature, 2012) have shown that Paneth cells can boost *Lgr5*+ CBC function in response to calorie restriction.

However, we do not claim that the loss of *Rfng* leads to depletion of Paneth cells. We find that in the absence of *Rfng*, the availability of DLL1 and DLL4 on the surface of Paneth cells reduces which results in reduced Notch activity in the neighbouring CBCs leading to their reduced self-renewal. These results are consistent with the observations of Pellegrinet et al., 2011 where they find that a complete loss of both DLL1 and DLL4 led to a depletion of *Lgr5*+ cells without any significant reduction in Paneth cell numbers.

To provide the additional verification requested, we performed Organoid Reconstitution Assay (Rodriguez-Coleman, MJ et al., Nature, 2017) to show that loss of *Rfng* in Paneth cells affects the *Lgr5+* CBCs. FACS sorted *Lgr5*-GFP cells were incubated with Paneth cells from wild type or *Rfng*null mice for 10 minutes at room temperature and plated in Matrigel. We find that the colony formation ability of *Lgr5+* CBCs incubated with Paneth cells lacking *Rfng* was significantly lower than the control (Figure 1—figure supplement 3A). It should be noted that not all CBCs associate with a Paneth cell during the incubation. Also the *Lgr5*+ stem cells divide and give rise to Paneth cells with *Rfng*. Hence the results of this assay are not as significant as that shown in Figure 1C.

As *Rfng*^-/-^ mice do not have *Lgr5*-GFP reporter allele, we were unable to isolate CBCs from them. Hence we are unable to show if loss of *Rfng* only in CBCs affects their colony formation ability. We added this caveat in the Discussion section. As *Rfng* is expressed mainly in the Paneth cells we feel that we would not see a marked difference in colony formation ability in this case.

We have replaced Figure 1F (now Figure 1G) to show *Lgr5* expression instead of *Olfm4* and *Ascl2,* as suggested.

Other comments:1) The authors state that "Notch signalling promotes self-renewal of CBCs" and reference the review by van der Flier and Clevers. In that review, inhibition of Notch leads to goblet differentiation, reduced proliferation, and fewer CBCs; however, this is often reversible. The authors should revise their statements accordingly.

We thank the reviewers for the comment. The statement has been revised to “Notch signalling is important for the maintenance of CBCs (Pellegrinet et al., 2011).”

2) The lysozyme expression appears very nuclear in Figure 1C. Conventional immunohistochemistry is required to assess the different cell populations clearly. The same nuclear localisation for the Lgr5-GFP is not in line with the previous published expression patterns and should be re-evaluated by the authors.

The staining appears nuclear in the figure shown owing to the smaller image size and the previous brightness and contrast settings. However, after we enlarged the image and increased the brightness and contrast, we can see that the staining is indeed cytoplasmic (Author response image 2). We have increased the size of the image in the main figure to avoid confusion. Also, our conclusions are mainly rely on the quantitative flow cytometric measurements.

**Author response image 2. respfig2:** Lysozyme (red) and GFP (green) staining in Paneth cells and CBCs respectively are cytoplasmic as expected. Figure 1D (previously Figure 1C) has been enlarged and brightness and contrast have been increased.

3) Given that goblet cell numbers vary along the length of the small intestine (SI) it is important the same parts of the small intestine are examined e.g. In some of the figures it appears that different parts of the SI have been analysed. The authors need to state which part has been analysed in their scoring and if not the same parts must be analysed.

The intestine was divided into 9 equal parts (from the proximal to distal end: duodenum-1, 2 and 3, jejunum-1, 2 and 3 and ileum-1, 2 and 3) while harvesting. Corresponding regions from control and mutated mice were compared. Scoring was mainly performed on ileal sections. Regions next to Peyer’s patches were excluded from the analysis.

4) To clearly show a reduction in Notch signalling activity after Lfng loss, the authors should examine the expression of Notch intra cellular domain (Nicd).

We thank the reviewers for this suggestion. We have stained the intestinal sections from *Lfng* null and control mice for NICD and observed a reduction in its expression in the crypts of *Lfng* null mice (Figure 5—figure supplement 2A).

5) Given that Notch signalling is not active in the villus, the function of Lfng in goblet cells on Notch signalling within these cells is confusing. A more likely explanation is that Lfng positive progenitors contribute to the described effects on colony formation as well as differentiation. The authors should clarify whether this progenitor population is controlling the detected effects rather the expression in goblet cells. For example, by Lfng-GFP sorting and exclusion of Muc2 positive cells, or Lfng-GFP and CD44 double positive cell sorting, followed by functional assays.

Goblet cells are known to express Notch ligands DLL1 and DLL4, however their role has not yet been fully understood. We find that *Lfng* in goblet cells can increase the cell-surface expression of these ligands. However, we are unable to ascertain the functional importance of *Lfng* in goblet cells.

Loss of *Lfng* leads to reduced Notch activity in the progenitors (Figure 5D) and this leads to an increase in the number of goblet cells (Pellegrinet et al., 2011), which is consistent with the reviewers’ explanation.

[Editors’ note: the author responses to the re-review follow. As indicated above, the manuscript was subsequently accepted for publication.]

[…] As you can from the verbatim comments, reviewer #3 was satisfied with your revisions. However, reviewer #2 felt that you had failed to address key points raised previously. In consultation, the Senior Editor agreed with reviewer #2 that your revisions were inadequate. Therefore we have agreed that the manuscript cannot be published in eLife.Reviewer #2:The authors have attempted to revise the manuscript in response to my comments. Unfortunately, the response to my comments has not really allayed my problems with the manuscript.The four points below have not, in my mind, been satisfactorily addressed and need to be.Point 1: Expression of fringe.The ISH data are not convincing for many reasons. First, RNAscope normally doesn't give dominantly signal in the nucleoli, based on the fact that mRNA is shuttled out of the nucleus after synthesis quickly.

We thank the reviewer for this observation. We agree that the result is puzzling as mRNA is shuttled out of the nucleus soon after synthesis. We consulted many experts to understand the contrariety. According to them, they have seen both (dominant signal from nucleus vs. cytoplasm) in many cases depending on the RNA species. The bright signals in the nuclei are likely to be pre-mRNA molecules or active transcription sites. Multiple nascent mRNA molecules are often found in a focus at transcription sites resulting in a bright spot (Raj A et al., Plos Biol, 2006). Mature mRNA molecules are often diffused in the cytoplasm and if the copy number of mRNA molecules is not very high it is possible that the only signals bright enough for detection would be from the transcription sites. We can see in (Figure 1, 2; Kwon S et al., Sci Rep, 2017) that the nuclear dots are quite brighter than those in the cytoplasm. We can see similar images in (Figure 1B, Bentovim L et al., Development, 2017; Figure S7, Levesque and Raj, Nat Meth, 2013). Accumulation of nascent mRNA in the nucleus, as the gene is transcribed, also results in a prominent nuclear signal (Lodish H et al., Mol. Cell Biol., 2000; Shermoen et al., Cell, 1991; Kwon S et al., Sci Rep, 2017). Stapel LC et al. (Development, 2016) also has shown that transcripts of certain genes are detected more often in the nucleus than in the cytoplasm.

That said, we have used RNAscope chromogenic assay for our studies. The chromogenic Fast Red signal obtained is also known to be fluorescent, and we acquired the data using a confocal fluorescent microscope. It should be noted that chromogenic RNAscope assay has more signal amplification steps when compared to the fluorescent assay. As a result, we may expect to see difference in the punctate signal in both cases. RNAscope chromogenic assays do often show a dominant signal in the nucleus. We have included a few snapshots of published images for the reviewers’ consideration to show that mRNA ISH using RNAscope can result in a dominant nuclear signal (please refer to Christensen AB et al, Mediators of Inflammation, 2015, Figure 4; Dominguez-Brauer C et al, Cell Stem Cell, 2016, Fig. S2D; Ziskin JL et al, Gut, 2012, Fig. 2F; Cheung EC et al, Genes and Dev., 2016, Suppl Fig. 2A; and Cammareri P et al, Cell Death Differ., 2017, Fig. 3A).

We counterstained the *Rfng*-ISH slides using haematoxylin and obtained a brightfield image (Author response image 3). As the tissue section is thick, was pre-treated with protease and haematoxylin was significantly diluted to prevent masking the ISH signal, stain appears weak. Nucleoli stain strongly with haematoxylin and we can see that ISH signal does not colocalise with bright blue spots. Although it is not possible to distinguish between heterochromatin and nucleolus in the image, it appears that ISH signal does not colocalise with either of them.

**Author response image 3. respfig3:** Representative image of RNAscope ISH for *Rfng* counterstained using haematoxylin on *Lgr5*-GFP mouse intestine. Arrows indicate regions with strong haematoxylin staining. Note that they do not overlap with the ISH signal. Scale bar represents 10μm.

As the above explanations only suggest but not completely confirm the legitimacy of the assay, we confirmed the specificity of the Rfng ISH probes by performing the assay on Rfng null intestines. We see no specific signal in absence of Rfng mRNA (Figure 1—figure supplement 1D, E). We performed in situ HCR for Rfng and observed nuclear dots. It should be noted that the assay is not as sensitive as the RNAscope chromogenic assay (Author response image 4).

We felt that an independent validation would boost the confidence in this observation. We analysed published microarray data (Sato T et al., Nature, 2011) on *Lgr5*+ CBCs and Paneth cells and found that Rfng is enriched in Paneth cells (Figure 1—figure supplement 1A).

**Author response image 4. respfig4:** Representative image of in situ HCR for *Rfng* on Lgr5-GFP mouse intestines shows signal primarily in the nuclei. Arrows point to nuclei with HCR signal. Dotted line separates the epithelium from the mesenchyme. Scale bar represents 10μm. Assay was performed as per manufacturer’s recommendation (Choi, Beck et al., 2014, Shah, Lubeck et al., 2016). V3.0 Probes for *Rfng* were purchased from Molecular Instruments. Tissues were pretreated using RNAscope reagents. Probes were hybridised to the section overnight at 40^0^C, washed using 5x SSCT buffer (5x SSC with 0.1% Tween-20) and signal was amplified overnight at 40^0^C. Slides were washed in SSCT, mounted and imaged using a confocal microscope. Spectral imaging and linear unmixing using Zen (Zeiss) software was performed to detect *Rfng* signal.

Additionally, why are Lysozyme positive cells in picture Figure 1B are not positive for Rfng? Quantification of Rnfg positive Paneth cells will be useful to understand the importance of Rnfg in Paneth cells.

Lysozyme positive cells in Figure 1B are positive for *Rfng*. As not all cells are centred on the same plane, the signal in any 2D image would vary from cell to cell. As it can been seen from the 3D rendering of the image for the reviewers’ consideration cited above (please refer to Christensen AB et al, Mediators of Inflammation, 2015, Figure 4; Dominguez-Brauer C et al, Cell Stem Cell, 2016, Fig. S2D; Ziskin JL et al, Gut, 2012, Fig. 2F; Cheung EC et al, Genes and Dev., 2016, Suppl Fig. 2A; and Cammareri P et al, Cell Death Differ., 2017, Fig. 3A), multiple punctate signals are present in different planes in different cells. And this would help us conclude that Lysozyme-expressing Paneth cells also express *Rfng*.

Second, cells outside the crypt are positive for Rnfg. How do the authors interpret these Rnfg-positive cells? And how do these cells contribute to the detected effects?

We thank the reviewer for this very interesting observation. We agree that some mesenchymal cells also express Rfng detectable by RNAscope assay. As Notch is a juxtacrine signalling pathway, we had confined the analysis to Paneth cells. Schröder et al., 2002 have shown, by RNA ISH, that the mesenchymal cells also express Rfng and this agrees with our images. We see signal in the mesenchymal cells by in situHCR (Author response image 4). We also observe that the mesenchymal cells also express *Dll1* (Figure 2—figure supplement 1). Further studies are needed to map the expression of all the Notch ligands in different mesenchymal cell types. This raises the possibility that the mesenchyme can also provide Notch ligands to the *Lgr5*+ CBCs in vivo. In case that is true, our proposed mechanism that RFNG promotes cell surface expression of DLL1 might be applicable to the mesenchymal cells too. Upon loss of *Rfng*, reduced DLL1 expression on the cell surface of Paneth cells and mesenchymal cells would result in reduced Notch activity in the CBCs, as observed. However, the epithelium is separated from the mesenchyme by a basement membrane. The efficacy of DLL mediated Notch signalling across the intestinal basement membrane would be an interesting scientific question by itself which we hope to address in our upcoming studies. We have included this limitation in the Discussion section to help readers interpret the data appropriately.

Point 2: Lysozyme expression.The enlarged figure is not sufficient and doesn't show cytoplasmic lysozyme expression at all. This technical issue needs to be corrected and much more convincing data needs to be provided. Data shown in Figure 1B show expression patterns for Lysosyme as expected and indicates that correct staining is possible.

We agree with the reviewer that lysozyme staining on organoids does not look as good as that on FFPE samples shown in Figure 1B. Whole organoids in Matrigel were fixed, stained and imaged in Figure 1D. The thickness of the tissue and the need to capture cells centred on different planes resulted in this appearance of Lysozyme signal. Similar staining has been reported previously (Basak et al., Cell Stem Cell, 2017). To avoid any confusion to the readers we have removed the image from the manuscript. Our conclusions are based on quantitative flow cytometry data in Figure 1E and F and hence would not be affected by removal of this image.

Point 4. To clearly show a reduction in Notch signalling activity after Lfng loss, the authors should examine the expression of Notch intra cellular domain (Nicd).It appears that the intensity of the IF presented in the revised version of the manuscript of NICD is much stronger rather than a change in NICD positive cells. The authors need to quantify this data for both points to be able to show the effect of Lfng loss. Furthermore, double staining of Lfng-GFP and NICD will help to understand the interaction of Lfng positive and Notch positive cells.

We thank the reviewers for suggesting these experiments. We quantified the fraction of nuclei positive for NICD in the crypts of *Lfng*^+/+^ and *Lfng*^-/-^ mouse intestines and found that *Lfng* loss indeed reduces the number of NICD+ nuclei (Figure 5—figure supplement 2A, B). We performed double staining of *Lfng*GFP and NICD and found that *Lfng*-GFP+ cells do not express NICD but are adjacent to NICD+ cells (Figure 4E).

Point 5. Given that Notch signalling is not active in the villus, the function of Lfng in goblet cells on Notch signalling within these cells is confusing.

Goblet cells have been shown previously (Shmizu et al., PeerJ, 2014) to express Notch ligands *Dll1* and *Dll4*, even though Notch signalling is not known to be active in the villus. We find that Lfng is also expressed by the *Dll1*+ *Dll4*+ goblet cells. We have found that LFNG, like RFNG, can promote DLL1 and DLL4 to their cell surface.

However, we agree with the reviewer that *Lfng* in goblet cells of the villi likely do not have a functional consequence in terms of Notch signalling or cell lineage determination. We further explored *Lfng* expression in the progenitor compartment, which is discussed in the following point.

Data which cell type is actually Lfng positive is not solid. Figure 4 doesn't show convincingly that Lfng is expressed in transit amplifying cells. This is of major importance since this would be the compartment that shows overlap with Notch activity and would explain the Lfng loss mediated effect on Notch signalling. Lfng positive Goblet cells in the villus can't explain this effect.

We thank the reviewer for encouraging us to probe further into the cells expressing *Lfng* in the crypts. We have since realised the importance of this experiment. We surprisingly found that the *Lfng*-GFP+ cells are not proliferating suggesting that they might be terminally differentiated (Figure 4A). As they are negative for activated Notch1 staining (Figure 4E), we hypothesised that they might be secretory cells. We find that they are positive for ChgA or Dclk1, which are markers for enteroendocrine or Tuft cells respectively (Figure 4B-D). Tuft, enteroendocrine cells and often goblet cells also are known to be present in the upper crypt (Gerbe et al., J Cell Biol, 2011; Barron et al., CMGH, 2017; VanDussen et al., 2012; Tian et al., Cell Rep, 2015). Enteroendocrine and goblet cells are known to express Notch ligand Dll1 (Shimizu et al., PeerJ, 2014, Van et al., 2012). These data suggest that LFNG in NICD- post-mitotic secretory cells of the upper crypt promotes Notch activity in the neighbouring enterocyte progenitors.